

# Unifying constructions of non-invertible symmetries

**Lakshya Bhardwaj[1], Sakura Schäfer-Nameki[1] and Apoorv Tiwari[2]**

**1** Mathematical Institute, University of Oxford, Andrew-Wiles Building,
Woodstock Road, Oxford, OX2 6GG, UK
**2** Department of Physics, KTH Royal Institute of Technology, Stockholm, Sweden

## Abstract

In the past year several constructions of non-invertible symmetries in Quantum Field Theory in $d \geq 3$ have appeared. In this paper we provide a unified perspective on these constructions. Central to this framework are so-called *theta defects*, which generalize the notion of theta-angles, and allow the construction of universal and non-universal topological symmetry defects. We complement this physical analysis by proposing a mathematical framework (based on higher-fusion categories) that converts the physical construction of non-invertible symmetries into a concrete computational scheme.

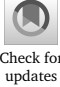

## 1 Introduction

Various distinct constructions of non-invertible symmetries have appeared in the literature in the last couple of years [1–9], with many followup applications [10–34]. This builds on

earlier work in 2d, where non-invertible symmetries have been studied in [35–47]. All of these non-invertible symmetries are built from invertible symmetries and dualities (suitably interpreted). This naturally makes one wonder, whether all these constructions are actually different manifestations of an underlying unified construction of non-invertible symmetries starting from symmetries that are invertible.

A summary of these distinct constructions is as follows:

1. Gauging outer-automorphisms (0-form symmetries): Field theoretically [1] and more systematically using a categorical formulation [5].

2. Gauging in the presence of mixed 't Hooft anomalies [2,5] by stacking TQFTs that carry a non-trivial anomaly themselves.

3. Gauging in theories with ABJ anomalies [6].

4. Dualities defects: [2,3].

5. Condensation defects: [4,6,7].

6. Stacking *G*-symmetric TQFTs (referred as *theta defects* here): [7].

7. Bimodule computations: [8].

In this paper we will show that these constructions are all special instances of a general, unified framework.

The key point is that defects after gauging an invertible symmetry Γ can be obtained from defects before gauging, possibly stacked with TQFTs. We call defects produced this way *(twisted) theta defects*. A defect *D* may lead to multiple defects after gauging, if there are multiple ways of 'coupling' *D* to Γ gauge fields in the bulk spacetime. On the other hand, a defect *D* may not lead to any defect after gauging if there is an obstruction to couple *D* to bulk Γ gauge fields (which may sometimes be curable by stacking a TQFT on top of the defect). These defects are a generalization of the theta[1] defects constructed in [7] by coupling decoupled TQFTs (of various codimensions) to bulk Γ gauge fields.

A mathematical study of different kinds of Γ-couplings for different kinds of defects at the level of symmetry categories[2] leads to familiar concepts in higher-category theory like higher-vector spaces and higher-representations of (higher-)groups. We cleanly describe the connection of the physical study of Γ-couplings to these categorical notions, along with the various subtleties related to Karoubi completions (physically known as condensations/gaugings) that arise for 3-categories and higher.

Non-invertible symmetries that arise from gauging invertible symmetries have been also referred to as non-intrinsic non-invertible symmetries [24]. They correspond to theories whose symmetry topological field theory (Symmetry TFT) [46, 51–55] is a (*d* + 1)-dimensional Dijkgraaf-Witten theory (possibly with twist). It will be interesting to make precise how intrinsic non-invertibles [27] fit into this framework. The Symmetry TFTs for such theories are obtained by a quotient of Dijkgraaf-Witten theories [24].

The paper has two main parts: a physics proposal for the construction of non-invertibles in section 2, and in section 3 a mathematical proposal for the relevant structures that are needed to capture the physical proposal, using concepts related to higher fusion categories. We conclude with a program for the classification of non-invertible symmetries in section 4.

---

[1]This terminology was not used in the paper [7].

[2]Let us briefly review the notion of 'symmetry category' following [5]: Given a *d*-dimensional QFT $\mathfrak{T}$ we can ask what symmetries it has. The general answer to this – to our current understanding – is that this is a set of topological defects of dimensions $0, \cdots, d-1$, which form a mathematical structure, a fusion $(d-1)$-category, known as the symmetry category of $\mathfrak{T}$. Fusion 2-categories were defined in [48] and, building upon it and [49], fusion higher-categories were given a definition in [50].

## 2 Non-invertible symmetries from invertibles: A unified perspective

The question we would like to answer in this paper is whether there is a unified construction of the non-invertible symmetry defects [1–9] in QFTs. In this section, we answer the question in the affirmative and present such a unified framework.

### 2.1 Theta angles

The construction is a generalization of the construction appearing in [7], which we now review. The construction of [7] can be understood as generalizing the notion of theta angle to higher-codimensions, and so we refer to the symmetries arising via this construction as *theta symmetries*.

**Higher-group symmetric QFTs.**  Consider a $d$-dimensional QFT $\mathfrak{T}$ with a non-anomalous higher-group symmetry[3] $\Gamma$. We can convert such a QFT to a '$\Gamma$-symmetric $d$-dimensional QFT' by choosing a gauge-invariant coupling[4] $\mathcal{S}$ of $\mathfrak{T}$ to background gauge fields for $\Gamma$. Such a coupling $\mathcal{S}$ allows one to define partition functions of $\mathfrak{T}$ in the presence of a background gauge field for $\Gamma$, and the fact that $\mathcal{S}$ is gauge-invariant means that these partition functions are the same for two background gauge fields related by a background gauge transformation. After choosing $\mathcal{S}$, we can label the $\Gamma$-symmetric $d$-dimensional QFT as $\mathfrak{T}_{\mathcal{S}}$. We also say that $\mathfrak{T}$ is the QFT underlying the $\Gamma$-symmetric QFT $\mathfrak{T}_{\mathcal{S}}$.

**Gauging a higher-group.**  Given a $\Gamma$-symmetric $d$-dimensional QFT $\mathfrak{T}_{\mathcal{S}}$, we can sum over the background gauge fields for $\Gamma$ consistently, or in other words we can gauge the $\Gamma$ symmetry. This produces a new $d$-dimensional QFT that we denote by $\mathfrak{T}_{\mathcal{S}}/\Gamma$. Note that, if $\mathcal{S}'$ is some other gauge invariant coupling of $\mathfrak{T}$ to background gauge fields for $\Gamma$, then the corresponding gauged theory $\mathfrak{T}_{\mathcal{S}'}/\Gamma$ is apriori different from $\mathfrak{T}_{\mathcal{S}}/\Gamma$, though there might exist a duality/isomorphism between the two for specific $\mathfrak{T}$.

**Product of higher-group symmetric QFTs.**  Now, given two $\Gamma$-symmetric $d$-dimensional QFTs $\mathfrak{T}_{\mathcal{S}}$ and $\mathfrak{T}'_{\mathcal{S}'}$, we can stack them to obtain a new $\Gamma$-symmetric $d$-dimensional QFT that can be denoted as

$$\mathfrak{T}_{\mathcal{S}} \otimes_{\Gamma} \mathfrak{T}'_{\mathcal{S}'}. \tag{1}$$

The $d$-dimensional QFT underlying $\mathfrak{T}_{\mathcal{S}} \otimes_{\Gamma} \mathfrak{T}'_{\mathcal{S}'}$ is the QFT $\mathfrak{T} \otimes \mathfrak{T}'$ obtained by stacking $\mathfrak{T}$ and $\mathfrak{T}'$. To promote the underlying QFT $\mathfrak{T} \otimes \mathfrak{T}'$ to a $\Gamma$-symmetric QFT $\mathfrak{T}_{\mathcal{S}} \otimes_{\Gamma} \mathfrak{T}'_{\mathcal{S}'}$, we need to first choose a $\Gamma$ symmetry of $\mathfrak{T} \otimes \mathfrak{T}'$ and then describe a coupling for this $\Gamma$ symmetry. The $\Gamma$ symmetry of $\mathfrak{T} \otimes \mathfrak{T}'$ is chosen to be the diagonal of the $\Gamma \times \Gamma$ symmetry of $\mathfrak{T} \otimes \mathfrak{T}'$ descending from $\Gamma$ symmetry of $\mathfrak{T}$ and $\Gamma$ symmetry of $\mathfrak{T}'$. The precise coupling of $\Gamma$ is then simply obtained by "adding" the couplings $\mathcal{S}$ and $\mathcal{S}'$.

**SPT phases protected by higher-group symmetry.**  The simplest $d$-dimensional QFT is the trivial $d$-dimensional TQFT $\mathfrak{I}$. The trivial theory $\mathfrak{I}$ is trivially symmetric under a non-anomalous higher-group symmetry $\Gamma$. But there can be various gauge-invariant couplings $\mathcal{S}$ of $\mathfrak{I}$ to background gauge fields for $\Gamma$. There is always a trivial coupling that we denote by $\mathcal{S}_0$.

---

[3]Throughout this paper, except for a short paragraph discussing the example of theta angle in $U(1)$ gauge theories, $\Gamma$ will be taken to be a completely finite higher-group, meaning that every $p$-form group $\Gamma^{(p)}$ inside $\Gamma$ is taken to be finite.

[4]We will give a precise mathematical definition of "coupling" in certain situations in next section. We hope that the usage of this word and concepts surrounding it in this section will be clear at an intuitive level.

The corresponding $\Gamma$-symmetric $d$-dimensional TQFTs $\mathfrak{I}_{\mathcal{S}}$ are often called as 'SPT phases protected by higher-group $\Gamma$'. That is, the underlying $d$-dimensional QFT for an SPT phase is the trivial theory $\mathfrak{I}$.

The SPT phases $\mathfrak{I}_{\mathcal{S}}$ form a group $\mathrm{SPT}_d^{\Gamma}$ under the above product operation (1). The identity element of the group is the SPT phase $\mathfrak{I}_{\mathcal{S}_0}$ which is also referred to as the trivial SPT phase.

**Monoid of higher-group symmetric QFTs.**    In fact, we have

$$\mathfrak{T}_{\mathcal{S}} \otimes_{\Gamma} \mathfrak{I}_{\mathcal{S}_0} = \mathfrak{T}_{\mathcal{S}}, \tag{2}$$

for any arbitrary $\Gamma$-symmetric $d$-dimensional QFT $\mathfrak{T}_{\mathcal{S}}$. Thus, $d$-dimensional $\Gamma$-symmetric QFTs form a monoid which the product being (1) and the identity element being $\mathfrak{I}_{\mathcal{S}_0}$.

**Theta angles.**    On the other hand, if we use a non-trivial SPT phase in the above stacking, we obtain

$$\mathfrak{T}_{\mathcal{S}} \otimes_{\Gamma} \mathfrak{I}_{\mathcal{S}''} = \mathfrak{T}_{\mathcal{S}'}, \tag{3}$$

where $\mathfrak{T}_{\mathcal{S}'}$ is a $\Gamma$-symmetric QFT obtained from $\mathfrak{T}$ by using a coupling $\mathcal{S}'$, which is closely related but different from the coupling $\mathcal{S}$.

As a consequence, the set

$$\mathfrak{T}_{\Gamma} = \{\mathfrak{T}_{\mathcal{S}}, \ \forall \, \mathcal{S}\}, \tag{4}$$

of all $\Gamma$-symmetric QFTs obtainable from the same underlying QFT $\mathfrak{T}$ admits a group action by the group $\mathrm{SPT}_d^{\Gamma}$. The action is free without any fixed points.

Let $\mathcal{O}_{\Gamma}^{\mathfrak{T}} \subseteq \mathfrak{T}_{\Gamma}$ be an orbit of the above group action. Pick two $\Gamma$-symmetric $d$-dimensional QFTs $\mathfrak{T}_{\mathcal{S}}$ and $\mathfrak{T}_{\mathcal{S}'}$ in the orbit $\mathcal{O}_{\Gamma}^{\mathfrak{T}}$. We have

$$\mathfrak{T}_{\mathcal{S}'} = \mathfrak{T}_{\mathcal{S}} \otimes_{\Gamma} \mathfrak{I}_{\mathcal{S}''}, \tag{5}$$

for a unique $\mathfrak{I}_{\mathcal{S}''} \in \mathrm{SPT}_d^{\Gamma}$. Correspondingly, it is often said that the $d$-dimensional QFT $\mathfrak{T}_{\mathcal{S}'}/\Gamma$ is related to the $d$-dimensional QFT $\mathfrak{T}_{\mathcal{S}}/\Gamma$ by the 'theta angle' $\mathfrak{I}_{\mathcal{S}''}$. See figure 1.

**Example: Theta angle in $U(1)$ gauge theory.**    Consider the 4d trivial theory $\mathfrak{I}$ and consider gauging its $U(1)$ 0-form symmetry. This leads to pure Maxwell theories $\mathfrak{T}_{\theta}$ in 4d, which is a family of theories differentiated by a circle valued parameter $\theta$, known as the theta angle. Two Maxwell theories $\mathfrak{T}_{\theta'}$ and $\mathfrak{T}_{\theta+\theta'}$ are related by a theta angle $\theta$, for which the corresponding $U(1)$ SPT phase has effective action

$$\theta \int F \wedge f \tag{6}$$

(here $F$ denotes the field strength for $U(1)$ 0-form background).

**Example: Discrete theta angle in $SO(3)$ gauge theory.**    It was pointed out in [56] that there are two versions of $SO(3)$ gauge theories in 4d, which are usually distinguished by labeling the gauge group as $SO(3)_{\pm}$ where the subscript $\pm$ labels a discrete $\mathbb{Z}_2$ valued theta angle.

This discrete theta angle is realized in terms of the above general construction as follows. Let $\mathfrak{T}$ be a 4d $SU(2)$ gauge theory whose matter content is such that the $\mathbb{Z}_2$ center of the $SU(2)$ gauge group survives as a $\mathbb{Z}_2$ 1-form symmetry of $\mathfrak{T}$. Now, there are (on a spin manifold) two 4d $\mathbb{Z}_2$ 1-form symmetric SPT phases, one whose effective action is trivial, and the other whose effective action is

$$\int \frac{\mathcal{P}(B_2)}{2}, \tag{7}$$

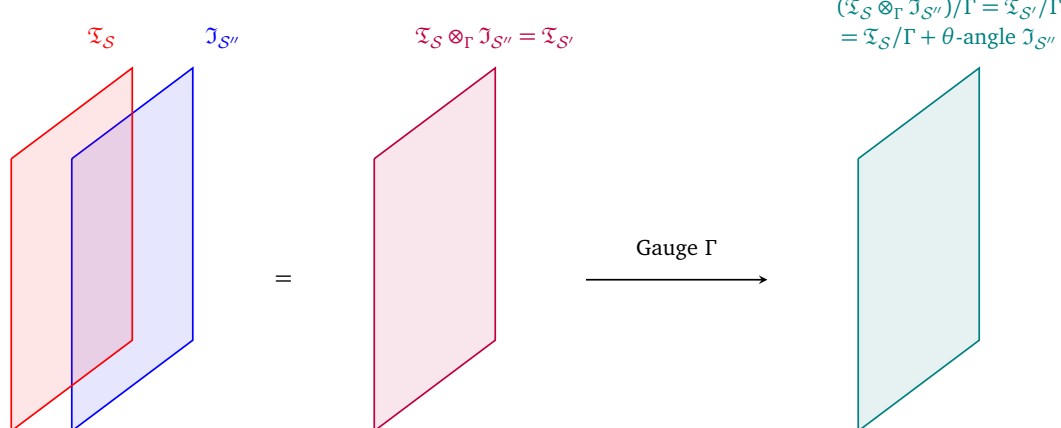

Figure 1: Stacking a $\Gamma$-symmetric $d$-dimensional QFT $\mathfrak{T}_{\mathcal{S}}$ with a $d$-dimensional SPT phase $\mathfrak{I}_{\mathcal{S}''}$ protected by $\Gamma$ higher-group symmetry leads to a new $\Gamma$-symmetric $d$-dimensional QFT $\mathfrak{T}_{\mathcal{S}} \otimes_{\Gamma} \mathfrak{I}_{\mathcal{S}''}$ which can be identified with the $d$-dimensional QFT $\mathfrak{T}_{\mathcal{S}'}$ obtained by choosing a different coupling $\mathcal{S}'$ of $\mathfrak{T}$ to $\Gamma$ backgrounds. Gauging further the $\Gamma$ symmetry leads to the $d$-dimensional QFT $(\mathfrak{T}_{\mathcal{S}} \otimes_{\Gamma} \mathfrak{I}_{\mathcal{S}''})/\Gamma = \mathfrak{T}_{\mathcal{S}'}/\Gamma$ which is also referred to as the $d$-dimensional QFT obtained by adding theta angle $\mathfrak{I}_{\mathcal{S}''}$ to the $d$-dimensional QFT $\mathfrak{T}_{\mathcal{S}}/\Gamma$ obtained by gauging the $\Gamma$ symmetry of $\mathfrak{T}_{\mathcal{S}}$.

where $\mathcal{P}(B_2)$ is the Pontryagin square of the 1-form symmetry background field $B_2$. Gauging the $\mathbb{Z}_2$ 1-form symmetry of $\mathfrak{T}$ converts the gauge group to $SU(2)/\mathbb{Z}_2 \cong SO(3)$, resulting in a 4d $SO(3)$ gauge theory. Depending on whether the invertible theory (7) is included or not in this gauging process, one either lands on the $SO(3)_+$ theory or the $SO(3)_-$ theory.

## 2.2 Theta symmetries

In this subsection we generalize the considerations of the previous subsection to (topological and non-topological) defects of a QFT.

**Defects from QFT stacking.** We begin our discussion with the general phenomenon that a $p$-dimensional QFT can always be treated as a $p$-dimensional defect of a $d$-dimensional QFT for $p < d$.

This can be understood by generalizing the stacking procedure above. We can stack a $p$-dimensional QFT $\mathsf{T}$ with a $d$-dimensional QFT $\mathfrak{T}$ for $p < d$ to obtain a $p$-dimensional defect $D_p^{(\mathsf{T})}$ of $\mathfrak{T}$. See figure 2. If $\mathsf{T}$ is a topological QFT, then $D_p^{(\mathsf{T})}$ is a topological defect of $\mathfrak{T}$. Otherwise, $D_p^{(\mathsf{T})}$ is a non-topological defect of $\mathfrak{T}$.

**Action of QFTs on defects.** We can actually generalize the stacking procedure to obtain an action of $p$-dimensional QFTs on $p$-dimensional defects of $\mathfrak{T}$. Stacking $\mathsf{T}$ with a $p$-dimensional defect $D_p$ of $\mathfrak{T}$ gives rise to a new $p$-dimensional defect

$$D_p^{(\mathsf{T})} \otimes D_p \,, \tag{8}$$

of $\mathfrak{T}$. See figure 3.

If $\mathsf{T}$ is a $p$-dimensional TQFT and $D_p$ is a topological defect, then $D_p^{(\mathsf{T})} \otimes D_p$ is another topological defect of $\mathfrak{T}$. This is related to the 'TQFT coefficients' of [4] as will become clear later.

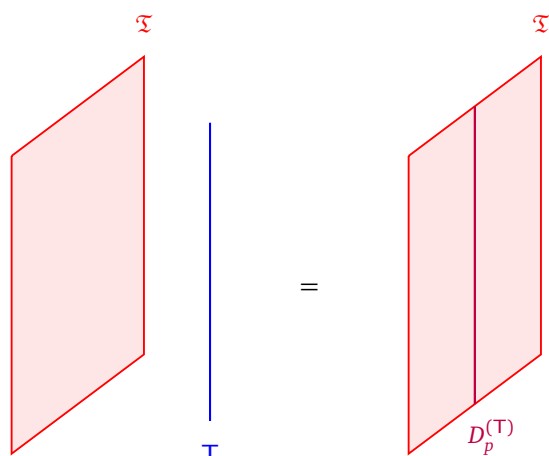

Figure 2: $\mathfrak{T}$ is a $d$-dimensional QFT and $\mathsf{T}$ is a $p$-dimensional QFT. Stacking $\mathsf{T}$ inside the spacetime occupied by $\mathfrak{T}$ produces a $p$-dimensional defect $D_p^{(\mathsf{T})}$ of $\mathfrak{T}$. If $\mathsf{T}$ is a TQFT, then $D_p^{(\mathsf{T})}$ is a topological defect of $\mathfrak{T}$.

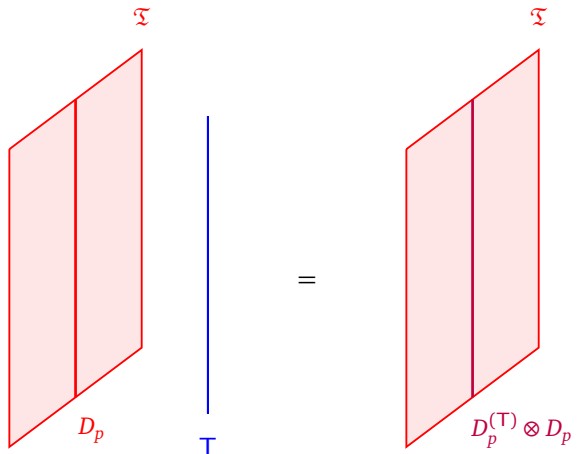

Figure 3: $\mathfrak{T}$ is a $d$-dimensional QFT, $\mathsf{T}$ is a $p$-dimensional QFT and $D_p$ is a $p$-dimensional defect of $\mathfrak{T}$. Stacking $\mathsf{T}$ inside the worldvolume occupied by $D_p$ produces a $p$-dimensional defect $D_p^{(\mathsf{T})} \otimes D_p$ of $\mathfrak{T}$. If $\mathsf{T}$ is a TQFT and $D_p$ is a topological defect, then $D_p^{(\mathsf{T})} \otimes D_p$ is a topological defect of $\mathfrak{T}$.

A special case arises if we take $\mathsf{T}$ to be a $p$-dimensional TQFT with $n$ trivial vacua and keep $D_p$ to be an arbitrary defect. Then, we have the equivalence

$$D_p^{(\mathsf{T})} \otimes D_p \cong n D_p \,, \tag{9}$$

where the right hand side denotes a direct sum of $n$ copies of $D_p$.

**Action of topological defects on general defects.** There is another action on general (topological or non-topological) $p$-dimensional defects of $\mathfrak{T}$ via *topological $p$-dimensional defects* of $\mathfrak{T}$. Stacking a $p$-dimensional topological defect $D_p$ of $\mathfrak{T}$ on top of a general $p$-dimensional defect $D_p'$ of $\mathfrak{T}$, we obtain a new $p$-dimensional defect

$$D_p \otimes D_p' \,, \tag{10}$$

of $\mathfrak{T}$. See figure 4.

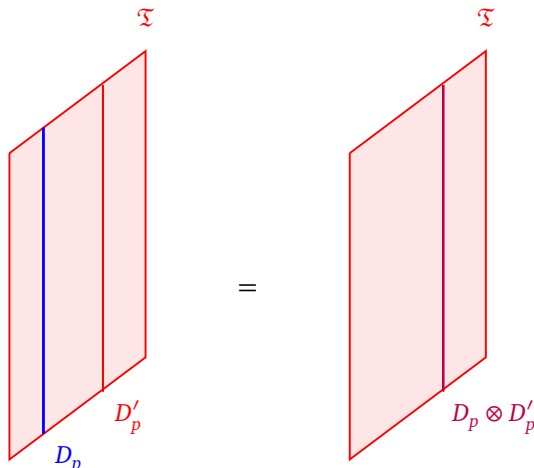

Figure 4: $\mathfrak{T}$ is a $d$-dimensional QFT, $D_p$ is a $p$-dimensional *topological* defect of $\mathfrak{T}$, $D_p'$ is a general, possibly non-topological, defect of $\mathfrak{T}$. Stacking $D_p$ inside the worldvolume occupied by $D_p'$ produces a general $p$-dimensional defect $D_p \otimes D_p'$ of $\mathfrak{T}$. If $D_p'$ is also topological, then $D_p \otimes D_p'$ is a topological defect of $\mathfrak{T}$.

In the special case that $D_p = D_p^{(\mathsf{T})}$ arises from a $p$-dimensional TQFT $\mathsf{T}$, we have defined two actions (8) and (10) of it on general $p$-dimensional defects of $\mathfrak{T}$. Both of these actions coincide.

**Fusion of topological defects.** If we take $D_p'$ to also be topological in (10), then the action of $D_p$ on $D_p'$ is simply the fusion of $D_p$ and $D_p'$.

**Higher-group symmetric defects.** Consider again a $d$-dimensional QFT $\mathfrak{T}$ with non-anomalous higher-group symmetry $\Gamma$. Let $\mathcal{S}$ be a gauge-invariant coupling of $\mathfrak{T}$ to $\Gamma$ background fields, and let $\mathfrak{T}_{\mathcal{S}}$ be the corresponding $\Gamma$-symmetric $d$-dimensional QFT. A $p$-dimensional (topological or non-topological) defect $D_p$ of $\mathfrak{T}$ can be converted into a '$\Gamma$-symmetric $p$-dimensional defect of $\mathfrak{T}_{\mathcal{S}}$' by choosing a gauge-invariant coupling $\mathcal{J}$ of $D_p$ to background gauge fields for $\Gamma$ living in the $d$-dimensional bulk spacetime. Combining such a defect coupling $\mathcal{J}$ with the bulk coupling $\mathcal{S}$ allows one to define correlation functions of $D_p$ in the presence of a background gauge field for $\Gamma$, and the fact that $\mathcal{J}$ is gauge-invariant means that these correlation functions are the same for two background gauge fields related by a background gauge transformation. Note that a choice of defect coupling $\mathcal{J}$ may be inconsistent for a choice of bulk coupling $\mathcal{S}$, but consistent for another bulk coupling $\mathcal{S}'$. After choosing $\mathcal{J}$, we can label the $\Gamma$-symmetric $p$-dimensional defect of $\mathfrak{T}_{\mathcal{S}}$ as $D_p^{(\mathcal{J})}$. We also say that $D_p$ is the defect underlying the $\Gamma$-symmetric defect $D_p^{(\mathcal{J})}$.

Note that there might not exist a gauge-invariant coupling $\mathcal{J}$ to $\Gamma$ backgrounds, or even a coupling afflicted with a 't Hooft anomaly, for some $p$-dimensional defects $D_p$ of $\mathfrak{T}$. On the other hand, for some $p$-dimensional defects $D_p$, there might exist multiple couplings $\mathcal{J}$ leading to multiple $\Gamma$-symmetric defects $D_p^{(\mathcal{J})}$.

The above description is a little quick for codimension-1 defects of $\mathfrak{T}$, i.e. for $p = d-1$. Such a defect partitions the $d$-dimensional spacetime occupied by $\mathfrak{T}$ into two parts, referred to as left and right in what follows. To convert a codimension-1 defect $D_{d-1}$ of $\mathfrak{T}$ into a $\Gamma$-symmetric codimension-1 defect of $\mathfrak{T}_{\mathcal{S}}$, we need to specify a coupling $\mathcal{J}_L$ of $D_{d-1}$ to background $\Gamma$ gauge fields living on the left-side of $D_{d-1}$ and a coupling $\mathcal{J}_R$ of $D_{d-1}$ to background $\Gamma$ gauge fields living on the right-side of $D_{d-1}$. Thus, the total coupling is $\mathcal{J} = (\mathcal{J}_L, \mathcal{J}_R)$ which we demand

to be gauge invariant under background gauge transformations on both left and right sides of $D_{d-1}$. In what follows, we will treat both cases $p < d-1$ and $p = d-1$ together, and in the latter case the coupling $\mathcal{J}$ will stand for the tuple $(\mathcal{J}_L, \mathcal{J}_R)$.

**Defects Surviving the Gauging.** As we discussed above, we can gauge the $\Gamma$ symmetry of $\mathfrak{T}$ with coupling $\mathcal{S}$ to obtain a $d$-dimensional QFT $\mathfrak{T}_{\mathcal{S}}/\Gamma$. A $\Gamma$-symmetric $p$-dimensional defect $D_p^{(\mathcal{J})}$ of $\mathfrak{T}_{\mathcal{S}}$ survives the gauging procedure due to gauge invariance of the coupling $\mathcal{J}$. Thus the defect $D_p^{(\mathcal{J})}$ becomes a $p$-dimensional defect $D_p^{(\mathcal{J})}/\Gamma$ of the gauged theory $\mathfrak{T}_{\mathcal{S}}/\Gamma$. Another $\Gamma$-symmetric $p$-dimensional defect $D_p^{(\mathcal{J}')}$ obtained by choosing a different coupling $\mathcal{J}'$ for the same underlying defect $D_p$ gives rise to a different $p$-dimensional defect $D_p^{(\mathcal{J}')}/\Gamma$ of the gauged theory $\mathfrak{T}_{\mathcal{S}}/\Gamma$, though for some specific $D_p$ the two defects $D_p^{(\mathcal{J})}/\Gamma$ and $D_p^{(\mathcal{J}')}/\Gamma$ may be dual/isomorphic.

If the defect $D_p$ of $\mathfrak{T}$ is topological, then the defect $D_p^{(\mathcal{J})}/\Gamma$ of $\mathfrak{T}_{\mathcal{S}}/\Gamma$ is also topological. In this case, the above procedure describes how to deduce the (invertible or non-invertible) symmetries of $\mathfrak{T}_{\mathcal{S}}/\Gamma$ from the information regarding the symmetries of $\mathfrak{T}$.

**$\Gamma$-symmetric defects by stacking $\Gamma$-symmetric QFTs.** Let $\mathsf{T}_{\mathcal{S}'}$ be a $\Gamma$-symmetric $p$-dimensional QFT with $\mathsf{T}$ being the underlying $p$-dimensional QFT. Then, rather similarly to figure 2, we can stack $\mathsf{T}_{\mathcal{S}'}$ in the spacetime occupied by $\mathfrak{T}_{\mathcal{S}}$ to obtain a $\Gamma$-symmetric $p$-dimensional defect

$$D_p^{(\mathsf{T}_{\mathcal{S}'})}, \tag{11}$$

of $\mathfrak{T}_{\mathcal{S}}$ whose underlying $p$-dimensional defect is $D_p^{(\mathsf{T})}$ discussed in figure 2. The coupling $\mathcal{J}$ for $D_p^{(\mathsf{T})}$ is obtained canonically from the coupling $\mathcal{S}'$, and hence we omit it.

If $\mathsf{T}$ is a TQFT, i.e. if $\mathsf{T}_{\mathcal{S}'}$ is a $\Gamma$-symmetric TQFT, then $D_p^{(\mathsf{T}_{\mathcal{S}'})}$ is a $p$-dimensional $\Gamma$-symmetric topological defect of $\mathfrak{T}_{\mathcal{S}}$.

**Theta symmetries.** Thus, in the gauged $d$-dimensional QFT $\mathfrak{T}_{\mathcal{S}}/\Gamma$, we obtain a universal sector

$$\left\{ D_p^{(\mathsf{T}_{\mathcal{S}'})}/\Gamma, \quad \forall\, \mathsf{T}_{\mathcal{S}'} \right\}, \tag{12}$$

of (generically non-topological) $p$-dimensional defects, parametrized by $\Gamma$-symmetric $p$-dimensional QFTs.

Restricting attention to those $\mathsf{T}$ that are TQFTs, we obtain a universal sector of generically non-invertible symmetries, parametrized by $\Gamma$-symmetric $p$-dimensional TQFTs, in any $d$-dimensional QFT $\mathfrak{T}_{\mathcal{S}}/\Gamma$ that can be obtained by gauging non-anomalous $\Gamma$ higher-group symmetry of another $d$-dimensional QFT $\mathfrak{T}$.

These symmetries were discussed in [7], and we refer to them as *theta symmetries* as their construction is a generalization of the construction of theta angles discussed above.

**Action of $\Gamma$-symmetric QFTs on $\Gamma$-symmetric defects.** We can generalize the above stacking procedure to obtain an action of $p$-dimensional $\Gamma$-symmetric QFTs on $p$-dimensional $\Gamma$-symmetric defects of the $d$-dimensional $\Gamma$-symmetric QFT $\mathfrak{T}_{\mathcal{S}}$. Stacking $\mathsf{T}_{\mathcal{S}'}$ with a $p$-dimensional $\Gamma$-symmetric defect $D_p^{(\mathcal{J})}$ of $\mathfrak{T}_{\mathcal{S}}$ gives rise to a new $\Gamma$-symmetric $p$-dimensional defect

$$D_p^{(\mathsf{T}_{\mathcal{S}'})} \otimes_\Gamma D_p^{(\mathcal{J})}, \tag{13}$$

of $\mathfrak{T}_{\mathcal{S}}$, whose underlying $p$-dimensional defect is

$$D_p^{(\mathsf{T})} \otimes D_p, \tag{14}$$

obtained by the action of the underlying QFT $\mathsf{T}$ of $\mathsf{T}_{\mathcal{S}'}$ on the underlying defect $D_p$ of $D_p^{(\mathcal{J})}$. If $\mathsf{T}$ is a $p$-dimensional TQFT and $D_p$ is a topological defect, then $D_p^{(\mathsf{T}_{\mathcal{S}'})} \otimes_\Gamma D_p$ is a $\Gamma$-symmetric $p$-dimensional topological defect of $\mathfrak{T}$.

**Action of $\Gamma$-symmetric topological defects on $\Gamma$-symmetric defects.** Stacking a $p$-dimensional $\Gamma$-symmetric topological defect $D_p^{(\mathcal{J})}$ of $\mathfrak{T}_{\mathcal{S}}$ on top of a general $p$-dimensional $\Gamma$-symmetric defect $D_p'^{(\mathcal{J}')}$ of $\mathfrak{T}_{\mathcal{S}}$, we obtain a new $p$-dimensional $\Gamma$-symmetric defect

$$D_p^{(\mathcal{J})} \otimes_\Gamma D_p'^{(\mathcal{J}')}, \tag{15}$$

of $\mathfrak{T}_{\mathcal{S}}$, whose underlying $p$-dimensional defect is

$$D_p \otimes D_p', \tag{16}$$

obtained by the action of underlying topological defect $D_p$ of $D_p^{(\mathcal{J})}$ on the underlying general defect $D_p'$ of $D_p'^{(\mathcal{J}')}$.

In the special case that $D_p^{(\mathcal{J})} = D_p^{(\mathsf{T}_{\mathcal{S}'})}$ arises from a $p$-dimensional $\Gamma$-symmetric TQFT $\mathsf{T}_{\mathcal{S}'}$, we have defined two actions (13) and (15) of it on general $\Gamma$-symmetric $p$-dimensional defects of $\mathfrak{T}_{\mathcal{S}}$. Both of these actions coincide.

**Fusion of topological defects after gauging.** Given two $\Gamma$-symmetric $p$-dimensional topological defects $D_p^{(\mathcal{J})}$ and $D_p'^{(\mathcal{J}')}$ of $\mathfrak{T}_{\mathcal{S}}$, we can act by one on the other to obtain another $\Gamma$-symmetric $p$-dimensional topological defect

$$D_p^{(\mathcal{J})} \otimes_\Gamma D_p'^{(\mathcal{J}')}, \tag{17}$$

whose underlying topological defect is the fused topological defect $D_p \otimes D_p'$
After gauging the $\Gamma$ symmetry, the above stacking descends to a fusion rule of $p$-dimensional topological defects of the $d$-dimensional QFT $\mathfrak{T}_{\mathcal{S}}/\Gamma$

$$\frac{D_p^{(\mathcal{J})}}{\Gamma} \otimes \frac{D_p'^{(\mathcal{J}')}}{\Gamma} = \frac{D_p^{(\mathcal{J})} \otimes_\Gamma D_p'^{(\mathcal{J}')}}{\Gamma}. \tag{18}$$

**Example: Dual higher-form symmetries.** Let $\mathfrak{T}$ be a $d$-dimensional QFT with a non-anomalous $p$-form symmetry described by an abelian group $\Gamma^{(p)}$. It is well-known that the $d$-dimensional QFT $\mathfrak{T}/\Gamma^{(p)}$ obtained after gauging[5] $\Gamma^{(p)}$ carries a 'dual' $(d-p-2)$-form symmetry described by the Pontryagin dual group $\widehat{\Gamma}^{(p)}$.

These dual symmetries are examples of theta symmetries: the topological defects generating the $(d-p-2)$-form symmetry of $\mathfrak{T}/\Gamma^{(p)}$ are $(p+1)$-dimensional and we label them as $D_{p+1}^{(\widehat{\gamma})}$ for elements $\widehat{\gamma} \in \widehat{\Gamma}^{(p)}$. The topological defect $D_{p+1}^{(\widehat{\gamma})}$ is the image of a $\Gamma^{(p)}$-symmetric $(p+1)$-dimensional topological defect $D_{p+1}^{(\mathfrak{I}_{\widehat{\gamma}})}$ of $\mathfrak{T}$ obtained by stacking a $(p+1)$-dimensional SPT phase protected by $\Gamma^{(p)}$ $p$-form symmetry $\mathfrak{I}_{\widehat{\gamma}}$. See figure 5. The effective action for the SPT phase $\mathfrak{I}_{\widehat{\gamma}}$ is

$$\int \widehat{\gamma}(B_{p+1}), \tag{19}$$

where $B_{p+1}$ is the $\Gamma^{(p)}$-valued $p$-form symmetry background field. The effective action is valued in $\mathbb{R}/\mathbb{Z}$ as it involves the canonical pairing $\widehat{\Gamma}^{(p)} \times \Gamma^{(p)} \to \mathbb{R}/\mathbb{Z}$. The corresponding theta defects

---

[5]We are suppressing the coupling $\mathcal{S}$ of $\mathfrak{T}$ to $\Gamma^{(p)}$ backgrounds as it does not play any role in what follows.

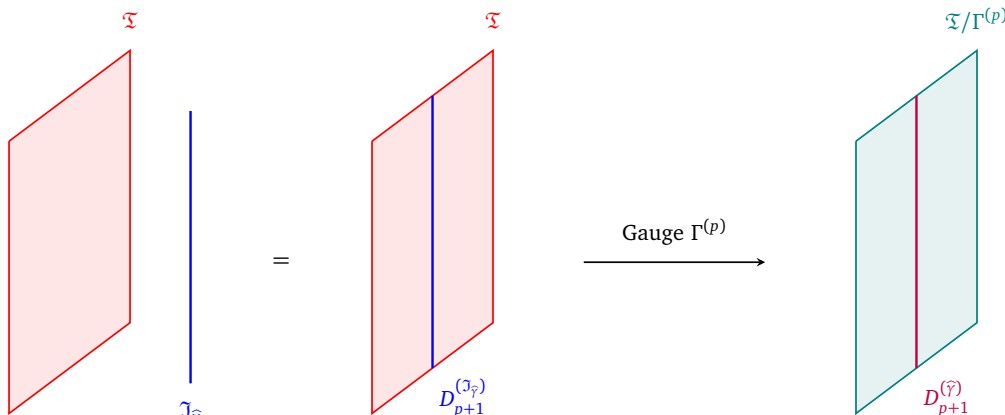

Figure 5: $\mathfrak{T}$ is a $d$-dimensional QFT with a non-anomalous $\Gamma^{(p)}$ $p$-form symmetry, and $\mathfrak{I}_{\widehat{\gamma}}$ is a $p+1$-dimensional SPT phase protected by $\Gamma^{(p)}$. Stacking $\mathfrak{I}_{\widehat{\gamma}}$ inside the spacetime occupied by $\mathfrak{T}$ produces a $\Gamma^{(p)}$-symmetric topological defect $D_{p+1}^{(\mathfrak{I}_{\widehat{\gamma}})}$ of $\mathfrak{T}$. Upon gauging $\Gamma^{(p)}$, we land on a $d$-dimensional QFT $\mathfrak{T}/\Gamma^{(p)}$ with a topological defect $D_{p+1}^{(\widehat{\gamma})}$. These $(p+1)$-dimensional topological defects generate a $\widehat{\Gamma}^{(p)}$ $(d-p-2)$-form symmetry of $\mathfrak{T}/\Gamma^{(p)}$.

$D_{p+1}^{(\widehat{\gamma})}$ in $\mathfrak{T}/\Gamma^{(p)}$ can also be interpreted as Wilson (hyper)surfaces/defects for the higher-form *dynamical* gauge field $B_{p+1}$ of $\mathfrak{T}/\Gamma^{(p)}$.

The fusion rules of theta defects $D_{p+1}^{(\widehat{\gamma})}$ are controlled by the group multiplication in $\widehat{\Gamma}^{(p)}$

$$D_{p+1}^{(\widehat{\gamma})} \otimes D_{p+1}^{(\widehat{\gamma}')} = D_{p+1}^{(\widehat{\gamma}+\widehat{\gamma}')}, \tag{20}$$

because SPT phases (19) form the group $\widehat{\Gamma}^{(p)}$ under stacking operation $\otimes_{\Gamma^{(p)}}$

$$\mathfrak{I}_{\widehat{\gamma}} \otimes_{\Gamma^{(p)}} \mathfrak{I}_{\widehat{\gamma}'} = \mathfrak{I}_{\widehat{\gamma}+\widehat{\gamma}'}. \tag{21}$$

**Example: Condensation surface defects for non-anomalous $(d-2)$-form symmetries.** Consider a $d$-dimensional QFT $\mathfrak{T}$ with a non-anomalous 0-form symmetry described by an abelian group $\Gamma^{(0)}$. As we discussed above, the $d$-dimensional QFT $\mathfrak{T}/\Gamma^{(0)}$ obtained after gauging $\Gamma^{(0)}$ (with any choice of coupling $\mathcal{S}$) carries a dual $(d-2)$-form symmetry generated by topological line defects valued in $\widehat{\Gamma}^{(0)}$. These lines can be condensed/gauged on a two-dimensional surface in the theory $\mathfrak{T}/\Gamma^{(0)}$ producing what are known as condensation surface defects, which in general generate non-invertible symmetries of $\mathfrak{T}/\Gamma^{(0)}$.

As described in [7] all such condensation surface defects can themselves be obtained as theta symmetries associated to the gauging procedure $\mathfrak{T} \to \mathfrak{T}/\Gamma^{(0)}$, by stacking $\mathfrak{T}$ with 2d TQFTs with $\Gamma^{(0)}$ non-anomalous 0-form symmetry, and then performing the $\Gamma^{(0)}$ gauging in the whole $d$-dimensional spacetime.

**Example: Condensation defects arising from duality defects.** Consider the example studied by the paper [2] involving a 4d spin-QFT $\mathfrak{T}$ with a $\mathbb{Z}_2^{(1)}$ 1-form symmetry and a $\mathbb{Z}_2^{(0)}$ 0-form with mixed 't Hooft anomaly

$$A_1 \cup \frac{\mathcal{P}(B_2)}{2}, \tag{22}$$

where $A_1$ is the background field for 0-form symmetry and $B_2$ is the background field for 1-form symmetry. The paper [2] constructs a non-invertible 3-dimensional topological defect

$D_3^{(S)}$ (known as a duality defect [3]) in the 4d theory $\mathfrak{T}/\mathbb{Z}_2^{(1)}$ obtained by gauging the $\mathbb{Z}_2^{(1)}$ 1-form symmetry, which has the fusion rule

$$D_3^{(S)} \otimes D_3^{(S)} \cong D_3^{(S\bar{S})}, \tag{23}$$

where $D_3^{(S\bar{S})}$ is a non-invertible 3-dimensional condensation defect that can be obtained by gauging the dual $\mathbb{Z}_2^{(2)}$ 2-form symmetry of $\mathfrak{T}/\mathbb{Z}_2^{(1)}$ along a 3-dimensional hypersurface with a discrete torsion specified by the non-trivial element of $H^3(B\mathbb{Z}_2^{(2)}, U(1)) \cong \mathbb{Z}_2$.

Below we describe how $D_3^{(S\bar{S})}$ can be understood as a theta defect. The defect $D_3^{(S)}$, on the other hand, will be discussed later as an example of a 'twisted' theta defect defined in the next subsection.

As remarked in the previous paragraph, the condensation defect $D_3^{(S\bar{S})}$ can be realized as a theta defect. This defect is obtained by stacking $\mathsf{T}_{S\bar{S}}$, which is the 3d Dijkgraaf-Witten TQFT based on a $\mathbb{Z}_2$ gauge group and a non-trivial twist, on $\mathfrak{T}$ before gauging the $\mathbb{Z}_2^{(1)}$ 1-form symmetry. The action of the Dijkgraaf-Witten theory can be written as

$$\int a_1 \cup \delta a_1 + a_1^3, \tag{24}$$

where $a_1$ is the $\mathbb{Z}_2$ gauge field. This theory $\mathsf{T}_{S\bar{S}}$ can be identified with the double semion model, which contains a bosonic $\mathbb{Z}_2$ line operator (namely the Deligne product of semion and anti-semion) generating a non-anomalous $\mathbb{Z}_2$ 1-form symmetry, which is used to convert the above double semion TQFT into a $\mathbb{Z}_2^{(1)}$ 1-form symmetric 3d TQFT, leading to the construction of the theta defect $D_3^{(S\bar{S})}$.

More generally one can consider theories with $\mathbb{Z}_N^{(1)}$ and $\mathbb{Z}_{2N}^{(0)}$ symmetries having similar mixed 't Hooft anomaly as (22), such as 4d pure Super-Yang-Mills [10, 19]. These theories also contain twisted theta defects whose fusion gives rise to condensation defects that can be understood as theta defects.

## 2.3 Twisted theta symmetries

We can incorporate a 'twist' in the construction of theta symmetries discussed above to construct a generalized version of theta symmetries that we refer to as *twisted theta symmetries*. Unlike theta symmetries, which are universal and exist in any theory that can be obtained by gauging an invertible higher-group symmetry of some other theory, the twisted theta symmetries are theory-dependent and hence non-universal.

Consider a situation in which we have a topological defect $D_p^{(m)}$ of $\mathfrak{T}$ that does not admit a gauge-invariant coupling $\mathcal{J}$ to background gauge fields for $\Gamma$ living in the bulk $d$-dimensional spacetime, for a specific choice of gauge-invariant bulk coupling $\mathcal{S}$. In other words, there is an obstruction for making $D_p^{(m)}$ $\Gamma$-symmetric. Let us also assume that $D_p^{(m)}$ is a *minimal* topological defect, that is

$$D_p^{(m)} \neq D_p^{(T)} \otimes D_p, \tag{25}$$

for any choices of $D_p^{(T)}$ and $D_p$, where $D_p^{(T)}$ is a $p$-dimensional defect of $\mathfrak{T}$ obtained by stacking some non-invertible[6] $p$-dimensional TQFT $\mathsf{T}$ and $D_p$ is some other $p$-dimensional topological defect of $\mathfrak{T}$.

---

[6]Note that this adjective is important for the definition to make sense, because every $p$-dimensional defect lies in a family of $p$-dimensional defects forming an orbit under action by $p$-dimensional invertible TQFTs.

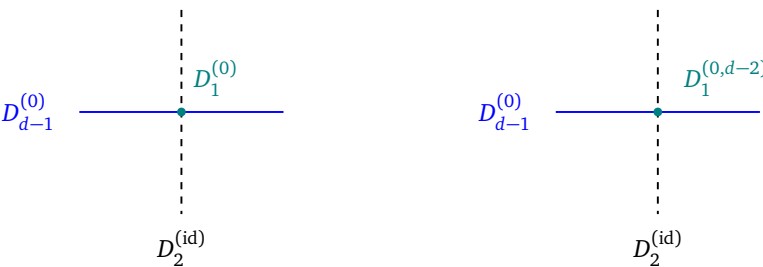

Figure 6: Two different gauge-invariant couplings $\mathcal{J}_0$ (on the left) and $\mathcal{J}_{0,d-2}$ (on the right) of the identity 2-dimensional defect $D_2^{(\mathrm{id})}$ (shown dashed) to $\mathbb{Z}_2^{(0)}$ 0-form symmetry backgrounds, distinguished by the choice of the line operator lying at the junction of $D_2^{(\mathrm{id})}$ and topological codimension-1 operator $D_{d-1}^{(0)}$ (shown in blue) generating $\mathbb{Z}_2^{(0)}$. See text for more details.

Now, suppose that we can stack a $p$-dimensional TQFT $\mathsf{T}$ on $D_p^{(m)}$ producing the $p$-dimensional topological defect

$$D_p^{(m,\mathsf{T})} := D_p^{(\mathsf{T})} \otimes D_p^{(m)} \,, \tag{26}$$

such that $D_p^{(m,\mathsf{T})}$ admits a gauge-invariant coupling $\mathcal{J}$ to bulk $\Gamma$ backgrounds, leading to a $\Gamma$-symmetric $p$-dimensional topological defect $D_p^{(m,\mathsf{T},\mathcal{J})}$ of $\mathfrak{T}_{\mathcal{S}}$. In other words, stacking by the TQFT $\mathsf{T}$ cures the obstruction for making $D_p^{(m)}$ $\Gamma$-symmetric.

We call the resulting $p$-dimensional topological defect

$$D_p^{(m,\mathsf{T},\mathcal{J})}/\Gamma \,, \tag{27}$$

of the gauged QFT $\mathfrak{T}_{\mathcal{S}}/\Gamma$ as a twisted theta defect, and the corresponding symmetry as *twisted theta symmetry*. We call $D_p^{(m)}$ as the underlying 'twist' and the TQFT $\mathsf{T}$ as the underlying 'stack' of the twisted theta defect $D_p^{(m,\mathsf{T},\mathcal{J})}$.

**Relationship between theta and twisted theta.** Note that a theta defect is a twisted theta defect with a trivial twist, i.e. the twist given by the identity $p$-dimensional defect $D_p^{(\mathrm{id})}$ of $\mathfrak{T}$; but converse is not true, as a twisted theta defect with a trivial twist might still involve a coupling $\mathcal{J}$ that is intrinsic to the QFT $\mathfrak{T}$ and cannot be decoupled to a coupling $\mathcal{J}$ for the stacked TQFT $\mathsf{T}$. An example is provided in the upcoming paper [9] which we reproduce in a generalized form below.

Consider a $d$-dimensional QFT $\mathfrak{T}$ with a $\mathbb{Z}_2^{(0)}$ 0-form symmetry generated by codimension-1 topological defect $D_{d-1}^{(0)}$ and $\mathbb{Z}_2^{(d-2)}$ $(d-2)$-form symmetry generated by a topological line defect $D_1^{(d-2)}$, without any 't Hooft anomalies for the two symmetries. There are then two possible couplings $\mathcal{J}$ for making the identity 2-dimensional defect $D_2^{(\mathrm{id})}$ symmetric under $\mathbb{Z}_2^{(0)}$. In one of them, labeled $\mathcal{J}_0$, the $\mathbb{Z}_2^{(0)}$ 0-form symmetry is implemented on $D_2^{(\mathrm{id})}$ by the identity line $D_1^{(0)}$ living on $D_{d-1}^{(0)}$. In the other, labeled $\mathcal{J}_{0,d-2}$, the $\mathbb{Z}_2^{(0)}$ 0-form symmetry is implemented on $D_2^{(\mathrm{id})}$ by the line operator $D_1^{(0,d-2)}$ living on $D_{d-1}^{(0)}$ obtained by stacking $D_1^{(d-2)}$ on top of the worldvolume of $D_{d-1}^{(0)}$. See figure 6.

The topological surface defect

$$D_2^{(\mathrm{id},\mathcal{J}_0)}/\mathbb{Z}_2^{(0)} \,, \tag{28}$$

in the QFT $\mathfrak{T}/\mathbb{Z}_2^{(0)}$ obtained after gauging $\mathbb{Z}_2^{(0)}$ 0-form symmetry of $\mathfrak{T}$ with a trivial choice of bulk coupling $\mathcal{S}$ is just the identity defect of $\mathfrak{T}/\mathbb{Z}_2^{(0)}$, which is a trivial theta defect. However,

on the other hand, the topological defect

$$D_2^{(\mathrm{id},\mathcal{J}_{0,d-2})}/\mathbb{Z}_2^{(0)}, \tag{29}$$

of $\mathfrak{T}/\mathbb{Z}_2^{(0)}$ is a twisted theta defect which is not a theta defect. It can be recognized as the condensation surface defect obtained by gauging the $\mathbb{Z}_2 \times \mathbb{Z}_2$ 1-form symmetry of $\mathfrak{T}/\mathbb{Z}_2^{(0)}$ on a two-dimensional worldvolume in spacetime along with a discrete torsion specified by the non-trivial element of $H^2(\mathbb{Z}_2 \times \mathbb{Z}_2, U(1)) \cong \mathbb{Z}_2$.

**Obstruction: Localized 't Hooft anomaly.**  It is interesting to classify the various kinds of obstructions for coupling $D_p^{(m)}$ consistently to a $\Gamma$ background. The most primitive type of obstruction is a 't Hooft anomaly for $\Gamma$ localized along the worldvolume of $D_p^{(m)}$. That is, there exists a coupling $\mathcal{J}$ of $D_p^{(m)}$ that allows one to define correlation functions of $D_p^{(m)}$ in the presence of $\Gamma$ backgrounds, when combined with a gauge-invariant bulk coupling $\mathcal{S}$. However, the coupling $\mathcal{J}$ is not gauge-invariant. That is, upon performing a background gauge transformation for $\Gamma$ background, the correlation functions of $D_p^{(m)}$ are transformed by a non-trivial phase. In other words, the coupling $\mathcal{J}$ suffers from a 't Hooft anomaly for $\Gamma$. This anomaly is localized along the defect $D_p^{(m)}$ because the bulk coupling $\mathcal{S}$ is gauge-invariant, that is partition functions of $\mathfrak{T}$ in the presence of $\Gamma$ backgrounds defined using the coupling $\mathcal{S}$ are invariant under background gauge transformations. We have a 't Hooft anomaly obstruction for coupling $D_p^{(m)}$ to $\Gamma$ backgrounds if we can only find a coupling $\mathcal{J}$ afflicted with a 't Hooft anomaly, but cannot find a gauge invariant coupling $\mathcal{J}$.

Now, consider a $p$-dimensional TQFT $\mathsf{T}$ with a coupling $\mathcal{S}'$ to the $\Gamma$ backgrounds afflicted with the inverse anomaly. The topological defect

$$D_p^{(m,\mathsf{T})} = D_p^{(\mathsf{T})} \otimes D_p^{(m)}, \tag{30}$$

then admits a gauge-invariant coupling "$\mathcal{J} + \mathcal{S}'$" to bulk $\Gamma$ backgrounds. The anomaly for the coupling $\mathcal{J}$ is canceled by the anomaly for the coupling $\mathcal{S}'$. We call the resulting $\Gamma$-symmetric $p$-dimensional topological defect of $\mathfrak{T}_{\mathcal{S}}$ as

$$D_p^{(m,\mathsf{T}_{\mathcal{S}'},\mathcal{J})}. \tag{31}$$

Gauging the $\Gamma$ symmetry, we obtain a $p$-dimensional twisted theta defect

$$D_p^{(m,\mathsf{T}_{\mathcal{S}'},\mathcal{J})}/\Gamma, \tag{32}$$

in the gauged QFT $\mathfrak{T}_{\mathcal{S}}/\Gamma$.

**Example: Duality defects (from mixed anomaly).**  This kind of obstruction appeared in the example studied by [2] that we discussed around equation (22). Let $D_3^{(0)}$ be the topological operator generating the $\mathbb{Z}_2^{(0)}$ 0-form symmetry. Then, the mixed 't Hooft anomaly (22) between the $\mathbb{Z}_2^{(0)}$ 0-form and $\mathbb{Z}_2^{(1)}$ 1-form symmetries descends to a 't Hooft anomaly

$$\frac{\mathcal{P}(B_2)}{2}, \tag{33}$$

for 1-form symmetry localized along the 3-dimensional worldvolume of $D_3^{(0)}$.

Now we look for a 3d TQFT with $\mathbb{Z}_2$ 1-form symmetry that carries the anomaly (33). One of the candidates, that was used in [2], is the semion model, or $U(1)_2$ Chern-Simons

theory $\mathsf{T_S}$.[7] This 3d TQFT carries a line operator, namely the semion, which generates a $\mathbb{Z}_2$ 1-form symmetry with anomaly (33).

Combining $\mathsf{T_S}$ with $D_3^{(0)}$, we obtain a $\mathbb{Z}_2^{(1)}$ 1-form symmetric 3d topological defect

$$D_3^{(0,\mathsf{T_S})} := D_3^{(\mathsf{T_S})} \otimes D_3^{(0)}, \tag{35}$$

of $\mathfrak{T}$. Gauging the $\mathbb{Z}_2^{(1)}$ 1-form symmetry, we obtain a 3d twisted theta defect

$$D_3^{(\mathsf{S})} = D_3^{(0,\mathsf{T_S})}/\mathbb{Z}_2^{(1)}, \tag{36}$$

in the 4d QFT $\mathfrak{T}/\mathbb{Z}_2^{(1)}$.

The fusion of this defect with itself was discussed in (22). This fusion can now be derived using (18). First of all, we could have used the 3d spin-TQFT $\mathsf{T_{\bar{S}}}$ given by the anti-semion model, for which the anti-semion line generates an anomalous $\mathbb{Z}_2$ 1-form symmetry, to construct another 3d twisted theta defect

$$D_3^{(\bar{\mathsf{S}})} = \frac{D_3^{(\mathsf{T_{\bar{S}}})} \otimes D_3^{(0)}}{\mathbb{Z}_2^{(1)}}. \tag{37}$$

Since $\mathsf{T_S}$ is isomorphic to $\mathsf{T_{\bar{S}}}$ as spin 3d TQFTs, the resulting twisted theta defects $D_3^{(\mathsf{S})}$ and $D_3^{(\bar{\mathsf{S}})}$ are also isomorphic (or in other words dual). Thus,

$$D_3^{(\mathsf{S})} \otimes D_3^{(\mathsf{S})} \cong D_3^{(\mathsf{S})} \otimes D_3^{(\bar{\mathsf{S}})}. \tag{38}$$

The right hand side is the twisted theta defect with trivial twist

$$D_3^{(0)} \otimes D_3^{(0)} = D_3^{(\mathrm{id})}, \tag{39}$$

and hence is a theta defect. The 3d TQFT used for stacking is the double semion model $\mathsf{T_{S\bar{S}}}$ discussed earlier since

$$\mathsf{T_S} \otimes \mathsf{T_{\bar{S}}} = \mathsf{T_{S\bar{S}}}. \tag{40}$$

The $\mathbb{Z}_2^{(1)}$ 1-form symmetry is implemented on $\mathsf{T_S}$ by the semion, on $\mathsf{T_{\bar{S}}}$ by the anti-semion, and hence must be implemented on $\mathsf{T_{S\bar{S}}}$ by the bosonic semion times anti-semion line. Thus, we see that the fusion is precisely the theta defect $D_3^{(S\bar{S})}$ discussed earlier

$$D_3^{(\mathsf{S})} \otimes D_3^{(\mathsf{S})} \cong D_3^{(S\bar{S})}. \tag{41}$$

Again there is a generalization to $\mathbb{Z}_N^{(1)}$ and $\mathbb{Z}_{2N}^{(0)}$ as in 4d $\mathcal{N} = 1$ Super-Yang-Mills. In this case the minimal 3d TQFT is $\mathcal{A}^{N,1} = U(1)_N$ [57], which has the opposite anomaly to the topological defect $D_3^{(0)}$ and the twisted theta defect in this case is

$$D_3^{(\mathsf{S})} = \frac{D_3^{(\mathcal{A}^{N,1})} \otimes D_3^{(0)}}{\mathbb{Z}_N^{(1)}}. \tag{42}$$

Let us note that many other examples of twisted theta defects generalizing the above construction of [2] have appeared in the literature since then. See [5, 10, 14, 58] for a sample of such works.

---

[7]To begin with, $\mathsf{S}$ is a 3d topological boundary condition of a 4d invertible TQFT $\mathfrak{I}_\mathsf{S}$ rather than a 3d TQFT. The partition function of $\mathfrak{I}_\mathcal{S}$ on a 4d manifold $M_4$ is

$$\exp\left(\frac{2\pi i \sigma(M_4)}{8}\right), \tag{34}$$

where $\sigma(M_4)$ is the signature of $M_4$. Consequently, $\mathfrak{I}_\mathsf{S}$ is invisible on a spin 4-manifold, because the signature of such a manifold is a multiple of 16. This fact allows us to treat $\mathsf{S}$ as a spin 3d TQFT, which is the right context here as the 4d QFT $\mathfrak{T}$ is a spin theory. We thank Jingxiang Wu for related discussions.

**Obstruction: Symmetry fractionalization.**    Above we saw an obstruction of coupling a defect $D_p^{(m)}$ to $\Gamma$ backgrounds, where a coupling $\mathcal{J}$ existed but it was not gauge-invariant leading to a 't Hooft anomaly for $\Gamma$ localized along $D_p^{(m)}$. We can also have "worse" obstructions, where even an anomalous coupling $\mathcal{J}$ cannot be found.

One of the simplest such obstructions is symmetry fractionalization of $\Gamma$ on the worldvolume of $D_p^{(m)}$, which is easiest to understand for a 0-form symmetry group $\Gamma^{(0)}$. This occurs when symmetry action of $\Gamma^{(0)}$ on $D_p$ does not close and in fact gives rise to a larger 0-form symmetry group $\Gamma_{\mathcal{E}}^{(0)}$ symmetry on $D_p$ which is an extension of the group $\Gamma^{(0)}$ leading to a short exact sequence

$$1 \to \Gamma_{D_p^{(m)}}^{(0)} \to \Gamma_{\mathcal{E}}^{(0)} \to \Gamma^{(0)} \to 1 \,, \tag{43}$$

with the key property being that the above sequence does not split. In such a situation, we say that $\Gamma^{(0)}$ 0-form symmetry is fractionalized to $\Gamma_{\mathcal{E}}^{(0)}$ 0-form symmetry on the worldvolume of $D_p^{(m)}$.

We can construct a twisted theta defect from $D_p^{(m)}$ if we can find a $p$-dimensional TQFT $\mathsf{T}$ such that

1. Stacking $\mathsf{T}$ on top of $D_p^{(m)}$ defractionalizes the $\Gamma_{\mathcal{E}}^{(0)}$ 0-form symmetry back to $\Gamma^{(0)}$ 0-form symmetry. That is, we can find a (possibly anomalous) coupling $\mathcal{J}$ of the $p$-dimensional defect $D_p^{(m,\mathsf{T})}$ to $\Gamma$ backgrounds.

2. The coupling $\mathcal{J}$ found in the previous step is actually non-anomalous/gauge-invariant.

If the above two conditions are satisfied, then we obtain a twisted theta defect

$$D_p^{(m,\mathsf{T},\mathcal{J})}/\Gamma \,, \tag{44}$$

in the gauged QFT $\mathfrak{T}_S/\Gamma$.

**Example: Gauging $\mathbb{Z}_4$ by gauging two $\mathbb{Z}_2$'s sequentially.**    We encounter an example of such an obstruction in the upcoming paper [9]. The context is the study of a $d$-dimensional QFT $\mathfrak{T}$ with $\mathbb{Z}_4$ non-anomalous 0-form symmetry. First gauge the $\mathbb{Z}_2$ subgroup of $\mathbb{Z}_4$ to pass on to the theory $\mathfrak{T}/\mathbb{Z}_2$. This theory has a residual $\mathbb{Z}_2^{(0)}$ 0-form symmetry and a dual $\mathbb{Z}_2^{(d-2)}$ $(d-2)$-form symmetry, with a mixed 't Hooft anomaly

$$B_{d-1} \cup A_1 \cup A_1 \,, \tag{45}$$

where $A_1$ is the $\mathbb{Z}_2^{(0)}$-valued background field for 0-form symmetry and $B_{d-1}$ is $\mathbb{Z}_2^{(d-2)}$-valued background field for $(d-2)$-form symmetry.

The $\mathbb{Z}_2^{(d-2)}$ symmetry is generated by topological line operators that can be condensed on a surface to give rise to a condensation surface defect that we label as $D_2^{(\mathbb{Z}_2)}$. We show in [9], that as a consequence of the mixed anomaly (45), a line operator $J$ living at the junction of $D_2^{(\mathbb{Z}_2)}$ and the generator $D_{d-1}^{(0)}$ for the $\mathbb{Z}_2^{(0)}$ 0-form symmetry has the property that

$$J^2 = P \,, \tag{46}$$

where $P$ is a non-trivial line operator living on $D_2^{(\mathbb{Z}_2)}$. See figure 7. Moreover, since

$$P^2 = 1 \,, \tag{47}$$

that is $P$ squares to the identity line on $D_2^{(\mathbb{Z}_2)}$, as shown in figure 7, we learn that

$$J^4 = 1 \,, \tag{48}$$

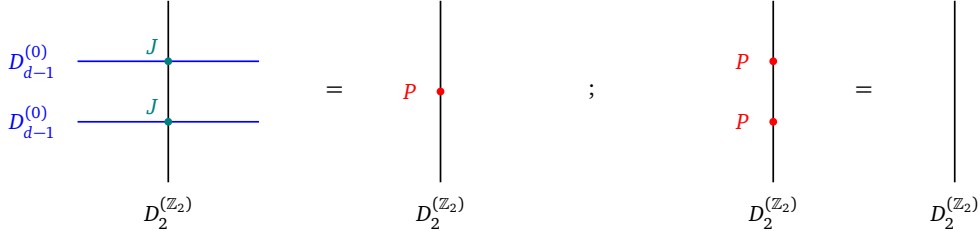

Figure 7: $J$ is a topological line defect arising at the junction of the codimension-1 topological defect $D_{d-1}^{(0)}$ generating the $\mathbb{Z}_2^{(0)}$ 0-form symmetry of $\mathfrak{T}/\mathbb{Z}_2$ and the dimension-2 condensation defect $D_2^{(\mathbb{Z}_2)}$ obtained by condensing the $\mathbb{Z}_2^{(d-2)}$ $(d-2)$-form symmetry. In other words, the bulk $\mathbb{Z}_2^{(0)}$ 0-form symmetry is generated on the surface defect $D_2^{(\mathbb{Z}_2)}$ by the line defect $J$. However, as shown in the figure, $J$ is not of order 2, because its square is a non-trivial line defect $P$ living on the surface $D_2^{(\mathbb{Z}_2)}$. Since $P$ is order 2, as shown in the figure, we learn that $J$ is order 4, and hence the $\mathbb{Z}_2^{(0)}$ 0-form symmetry in the bulk fractionalizes to $\mathbb{Z}_4$ symmetry on the surface $D_2^{(\mathbb{Z}_2)}$.

implying that the $\mathbb{Z}_2^{(0)}$ 0-form symmetry fractionalizes to a $\mathbb{Z}_4$ 0-form symmetry on the world-volume of $D_2^{(\mathbb{Z}_2)}$. Consequently, $D_2^{(\mathbb{Z}_2)}$ does not give rise to a topological surface defect of the theory $\mathfrak{T}/\mathbb{Z}_4$ obtained from $\mathfrak{T}/\mathbb{Z}_2$ by gauging its residual $\mathbb{Z}_2^{(0)}$ 0-form symmetry.

However the symmetry can be defractionalized on the surface defect

$$D_2^{(\mathbb{Z}_2,\mathsf{T})} := \mathsf{T} \otimes D_2^{(\mathbb{Z}_2)} \cong 2D_2^{(\mathbb{Z}_2)}, \tag{49}$$

where $\mathsf{T}$ is a 2d TQFT with two trivial vacua. The line operators on $D_2^{(\mathbb{Z}_2,\mathsf{T})} \cong 2D_2^{(\mathbb{Z}_2)}$ are $2 \times 2$ matrices with elements being line operators on $D_2^{(\mathbb{Z}_2)}$. The $(ij)$-th entry in the matrix is a line operator from the $i$-th copy of $D_2^{(\mathbb{Z}_2)}$ to the $j$-th copy of $D_2^{(\mathbb{Z}_2)}$. Then the matrix

$$\mathcal{J} := \begin{pmatrix} 0 & J \\ J^3 & 0 \end{pmatrix} = \begin{pmatrix} 0 & J \\ P \otimes J & 0 \end{pmatrix}, \tag{50}$$

provides a line operator living at the junction of $D_2^{(\mathbb{Z}_2,\mathsf{T})}$ and $D_{d-1}$, which squares to identity

$$\mathcal{J}^2 = 1, \tag{51}$$

and hence it is possible to have non-fractionalized $\mathbb{Z}_2^{(0)}$ 0-form symmetry on the worldvolume of $D_2^{(\mathbb{Z}_2,\mathsf{T})}$. In fact, $\mathcal{J}$ provides a non-anomalous coupling giving rise to a twisted theta defect $D_2^{(\mathbb{Z}_2,\mathsf{T},\mathcal{J})}/\mathbb{Z}_2^{(0)}$ of the theory $\mathfrak{T}/\mathbb{Z}_4$, which can also be recognized as the surface defect obtained by condensing the topological lines generating the $\mathbb{Z}_4^{(d-2)}$ $(d-2)$-form symmetry of the theory $\mathfrak{T}/\mathbb{Z}_4$.

## 2.4 Symmetries from topological interfaces

**Interfaces.** Interfaces are (topological or non-topological) $(d-1)$-dimensional defects living between two $d$-dimensional QFTs $\mathfrak{T}^{(L)}$ and $\mathfrak{T}^{(R)}$. We will say that an interface $D_{d-1}$ is 'from $\mathfrak{T}^{(L)}$ to $\mathfrak{T}^{(R)}$', if $\mathfrak{T}^{(L)}$ lives on the left side of $I_{d-1}$ and $\mathfrak{T}^{(R)}$ lives on the right side of $D_{d-1}$.

If $\mathfrak{T}^{(L)} = \mathfrak{T}^{(R)} = \mathfrak{T}$, then the interfaces between $\mathfrak{T}^{(L)}$ to $\mathfrak{T}^{(R)}$ are the same as codimension-1 defects of the $d$-dimensional QFT $\mathfrak{T}$.

**Action of QFTs on interfaces.** We can stack $(d-1)$-dimensional QFTs on interfaces from $\mathfrak{T}^{(L)}$ to $\mathfrak{T}^{(R)}$ to obtain new interfaces from $\mathfrak{T}^{(L)}$ to $\mathfrak{T}^{(R)}$. Stacking a $(d-1)$-dimensional QFT $\mathsf{T}$ with an interface $I_{d-1}$ creates a new interface that we denote as

$$I_{d-1}^{(\mathsf{T})}. \tag{52}$$

This process is rather similar to the one shown in figure 3. If $\mathsf{T}$ is a $(d-1)$-dimensional TQFT and $I_{d-1}$ is a topological interface, then $I_{d-1}^{(\mathsf{T})}$ is another topological interface.

As for defects, a special case arises if we take $\mathsf{T}$ to be a $(d-1)$-dimensional TQFT with $n$ trivial vacua and keep $I_{d-1}$ to be an arbitrary defect. Then, we have the equivalence

$$I_{d-1}^{(\mathsf{T})} \cong n I_{d-1}, \tag{53}$$

where the right hand side denotes a direct sum of $n$ copies of $I_{d-1}$.

**Actions of topological interfaces on general interfaces.** Consider a topological interface $I_{d-1}$ from $\mathfrak{T}_1$ to $\mathfrak{T}_2$, and a general (topological or non-topological) interface $I'_{d-1}$ from $\mathfrak{T}_2$ to $\mathfrak{T}_3$. We can act from the left by $I_{d-1}$ on $I'_{d-1}$ to obtain an interface

$$I_{d-1} \otimes I'_{d-1}, \tag{54}$$

from $\mathfrak{T}_1$ to $\mathfrak{T}_3$. If $I'_{d-1}$ is topological, then $I_{d-1} \otimes I'_{d-1}$ is also topological. In this case, the topological interface $I_{d-1} \otimes I'_{d-1}$ is referred to the fusion of the topological interfaces $I_{d-1}$ and $I'_{d-1}$. If $\mathfrak{T}_1 = \mathfrak{T}_2 \equiv \mathfrak{T}$, then the above is a left action of topological defects of $\mathfrak{T}$ on interfaces from $\mathfrak{T}$ to $\mathfrak{T}_3$.

Similarly, if $I_{d-1}$ is a general interface from $\mathfrak{T}_1$ to $\mathfrak{T}_2$ and $I'_{d-1}$ is a topological interface from $\mathfrak{T}_2$ to $\mathfrak{T}_3$, we can act from the right by $I'_{d-1}$ on $I_{d-1}$ to obtain an interface

$$I_{d-1} \otimes I'_{d-1}, \tag{55}$$

from $\mathfrak{T}_1$ to $\mathfrak{T}_3$. If $I_{d-1}$ is topological, then $I_{d-1} \otimes I'_{d-1}$ is also topological. In this case, the topological interface $I_{d-1} \otimes I'_{d-1}$ is referred to the fusion of the topological interfaces $I_{d-1}$ and $I'_{d-1}$. If $\mathfrak{T}_2 = \mathfrak{T}_3 \equiv \mathfrak{T}$, then the above is a right action of topological defects of $\mathfrak{T}$ on interfaces from $\mathfrak{T}_1$ to $\mathfrak{T}$.

These actions are rather similar to the action shown in figure 4.

Consider a $(d-1)$-dimensional TQFT $\mathsf{T}$. Then we have defined three actions of it on a general interface $I_{d-1}$. First we can directly stack it on top of $I_{d-1}$ to obtain the interface $I_{d-1}^{(\mathsf{T})}$ discussed in (52). Second, we can first stack $\mathsf{T}$ along $\mathfrak{T}^{(L)}$ to obtain a codimension-1 topological defect $D_{d-1}^{(L,\mathsf{T})}$ of $\mathfrak{T}^{(L)}$, which we can act on $I_{d-1}$ from the left to obtain the interface $D_{d-1}^{(L,\mathsf{T})} \otimes I_{d-1}$ discussed in (54). Third, we can first stack $\mathsf{T}$ along $\mathfrak{T}^{(R)}$ to obtain a codimension-1 topological defect $D_{d-1}^{(R,\mathsf{T})}$ of $\mathfrak{T}^{(R)}$, which we can act on $I_{d-1}$ from the right to obtain the interface $I_{d-1} \otimes D_{d-1}^{(R,\mathsf{T})}$ discussed in (55). All these processes lead to the same interface, i.e. we have the equalities

$$I_{d-1}^{(\mathsf{T})} = D_{d-1}^{(L,\mathsf{T})} \otimes I_{d-1} = I_{d-1} \otimes D_{d-1}^{(R,\mathsf{T})}. \tag{56}$$

**Higher-group symmetric interfaces.** For $i \in \{L, R\}$, let $\mathfrak{T}^{(i)}$ have a non-anomalous $\Gamma_i$ higher-group symmetry. Choose a gauge-invariant coupling $\mathcal{S}_i$ of $\mathfrak{T}^{(i)}$ to background gauge fields for $\Gamma_i$.

An interface $I_{d-1}$ from $\mathfrak{T}^{(L)}$ to $\mathfrak{T}^{(R)}$ can be converted into a $\Gamma_L$-symmetric interface $I_{d-1}^{(\mathcal{J}_L)}$ from the $\Gamma_L$-symmetric $d$-dimensional QFT $\mathfrak{T}_{\mathcal{S}_L}^{(L)}$ to the $d$-dimensional QFT $\mathfrak{T}^{(R)}$ by providing a

gauge-invariant coupling $\mathcal{J}_L$ of $I_{d-1}$ to background gauge fields for $\Gamma_L$ living on the left of $I_{d-1}$. Such a coupling $\mathcal{J}_L$ combined with the coupling $\mathcal{S}$ allows us to define correlation functions involving $I_{d-1}$, $\mathfrak{T}^{(L)}$ and $\mathfrak{T}^{(R)}$ in the presence of background fields for $\Gamma_L$ living on the portion of spacetime occupied by $\mathfrak{T}^{(L)}$. The gauge invariance of the coupling $\mathcal{J}_L$ translates to the fact that these correlation functions are left invariant if we perform background gauge transformations on $\Gamma_L$ background gauge fields.

Similarly, an interface $I_{d-1}$ from $\mathfrak{T}^{(L)}$ to $\mathfrak{T}^{(R)}$ can be converted into a $\Gamma_R$-symmetric interface $I_{d-1}^{(\mathcal{J}_R)}$ from the $d$-dimensional QFT $\mathfrak{T}^{(L)}$ to the $\Gamma_R$-symmetric $d$-dimensional QFT $\mathfrak{T}^{(R)}_{\mathcal{S}_R}$ by providing a gauge-invariant coupling $\mathcal{J}_R$ of $I_{d-1}$ to background gauge fields for $\Gamma_R$ living on the right of $I_{d-1}$. Such a coupling $\mathcal{J}_R$ combined with the coupling $\mathcal{S}$ allows us to define correlation functions involving $I_{d-1}$, $\mathfrak{T}^{(L)}$ and $\mathfrak{T}^{(R)}$ in the presence of background fields for $\Gamma_R$ living on the portion of spacetime occupied by $\mathfrak{T}^{(R)}$. The gauge invariance of the coupling $\mathcal{J}_R$ translates to the fact that these correlation functions are left invariant if we perform background gauge transformations on $\Gamma_R$ background gauge fields.

Combining the above two, an interface $I_{d-1}$ from $\mathfrak{T}^{(L)}$ to $\mathfrak{T}^{(R)}$ can be converted into a $(\Gamma_L, \Gamma_R)$-symmetric interface $I_{d-1}^{(\mathcal{J}_L, \mathcal{J}_R)}$ from the $\Gamma_L$-symmetric $d$-dimensional QFT $\mathfrak{T}^{(L)}_{\mathcal{S}_L}$ to the $\Gamma_R$-symmetric $d$-dimensional QFT $\mathfrak{T}^{(R)}$ by providing a coupling $\mathcal{J}_L$ of $I_{d-1}$ to background gauge fields for $\Gamma_L$ living on the left of $I_{d-1}$ and a coupling $\mathcal{J}_R$ of $I_{d-1}$ to background gauge fields for $\Gamma_R$ living on the right of $I_{d-1}$, such that the combined coupling $(\mathcal{J}_L, \mathcal{J}_R)$ is gauge-invariant.

The interface couplings $(\mathcal{J}_L, \mathcal{J}_R)$ combined with the bulk couplings $(\mathcal{S}_L, \mathcal{S}_R)$ allow us to define correlation functions involving $I_{d-1}$, $\mathfrak{T}^{(L)}$ and $\mathfrak{T}^{(R)}$ in the presence of background fields for $\Gamma_L$ living on the portion of spacetime occupied by $\mathfrak{T}^{(L)}$ and background fields for $\Gamma_R$ living on the portion of spacetime occupied by $\mathfrak{T}^{(R)}$. The gauge invariance of the coupling $(\mathcal{J}_L, \mathcal{J}_R)$ translates to the fact that these correlation functions are left invariant if we perform background gauge transformations on both $\Gamma_L$ and $\Gamma_R$ valued background gauge fields.

**Interfaces surviving gauging.** A $\Gamma_L$-symmetric interface $I_{d-1}^{(\mathcal{J}_L)}$ survives the procedure of gauging $\Gamma_L$ symmetry of $\mathfrak{T}^{(L)}_{\mathcal{S}_L}$ leading to an interface

$$I_{d-1}^{(\mathcal{J}_L)}/\Gamma_L\,, \tag{57}$$

from the $d$-dimensional QFT $\mathfrak{T}^{(L)}_{\mathcal{S}_L}/\Gamma_L$ to the $d$-dimensional QFT $\mathfrak{T}^{(R)}$.

Similarly, a $\Gamma_R$-symmetric interface $I_{d-1}^{(\mathcal{J}_R)}$ survives the procedure of gauging $\Gamma_R$ symmetry of $\mathfrak{T}^{(R)}_{\mathcal{S}_R}$ leading to an interface

$$I_{d-1}^{(\mathcal{J}_R)}/\Gamma_R\,, \tag{58}$$

from the $d$-dimensional QFT $\mathfrak{T}^{(L)}$ to the $d$-dimensional QFT $\mathfrak{T}^{(R)}_{\mathcal{S}_R}/\Gamma_R$.

Finally, a $(\Gamma_L, \Gamma_R)$-symmetric interface $I_{d-1}^{(\mathcal{J}_L, \mathcal{J}_R)}$ survives the procedure of gauging both $\Gamma_L$ symmetry of $\mathfrak{T}^{(L)}_{\mathcal{S}_L}$ and $\Gamma_R$ symmetry of $\mathfrak{T}^{(R)}_{\mathcal{S}_R}$, leading to an interface

$$I_{d-1}^{(\mathcal{J}_L, \mathcal{J}_R)}/(\Gamma_L, \Gamma_R)\,, \tag{59}$$

from the $d$-dimensional QFT $\mathfrak{T}^{(L)}_{\mathcal{S}_L}/\Gamma_L$ to the $d$-dimensional QFT $\mathfrak{T}^{(R)}_{\mathcal{S}_R}/\Gamma_R$.

For $\mathfrak{T}_L = \mathfrak{T}_R \equiv \mathfrak{T}$, $\Gamma_L = \Gamma_R \equiv \Gamma$ and $\mathcal{S}_L = \mathcal{S}_R \equiv \mathcal{S}$, the interface $I_{d-1}^{(\mathcal{J}_L, \mathcal{J}_R)}/(\Gamma_L, \Gamma_R)$ is the codimension-1 defect $I_{d-1}^{(\mathcal{J})}/\Gamma$ of the $d$-dimensional QFT $\mathfrak{T}_{\mathcal{S}}/\Gamma$ obtained from the codimension-1 defect $I_{d-1}$ of $\mathfrak{T}$ with coupling $\mathcal{J} \equiv (\mathcal{J}_L, \mathcal{J}_R)$.

Moving forward, we treat a $\Gamma_L$-symmetric interface $I_{d-1}^{(\mathcal{J}_L)}$ as a $(\Gamma_L, \Gamma_R)$-symmetric interface $I_{d-1}^{(\mathcal{J}_L, 0)}$ for $\Gamma_R = \mathcal{S}_R = \mathcal{J}_R = 0$. Similarly, we treat a $\Gamma_R$-symmetric interface $I_{d-1}^{(\mathcal{J}_R)}$ as a $(\Gamma_L, \Gamma_R)$-symmetric interface $I_{d-1}^{(0, \mathcal{J}_R)}$ for $\Gamma_L = \mathcal{S}_L = \mathcal{J}_L = 0$.

**Actions of topological symmetric interfaces on general symmetric interfaces.** A topological $(\Gamma_1, \Gamma_2)$-symmetric interface $I_{d-1}^{(\mathcal{J}_1, \mathcal{J}_2)}$ from $\mathfrak{T}_{\mathcal{S}_1}^{(1)}$ to $\mathfrak{T}_{\mathcal{S}_2}^{(2)}$ can act from the left on a general (topological or non-topological) $(\Gamma_2, \Gamma_3)$-symmetric interface $I'^{(\mathcal{J}'_2, \mathcal{J}'_3)}_{d-1}$ from $\mathfrak{T}_{\mathcal{S}_2}^{(2)}$ to $\mathfrak{T}_{\mathcal{S}_3}^{(3)}$ to give rise to a $(\Gamma_1, \Gamma_3)$-symmetric interface

$$I_{d-1}^{(\mathcal{J}_1, \mathcal{J}_2)} \otimes_{\Gamma_2} I'^{(\mathcal{J}'_2, \mathcal{J}'_3)}_{d-1}, \tag{60}$$

from $\mathfrak{T}_{\mathcal{S}_1}^{(1)}$ to $\mathfrak{T}_{\mathcal{S}_3}^{(3)}$, whose underlying interface is

$$I_{d-1} \otimes I'_{d-1}, \tag{61}$$

from $\mathfrak{T}_1$ to $\mathfrak{T}_3$. The coupling of $I_{d-1}^{(\mathcal{J}_1, \mathcal{J}_2)} \otimes_{\Gamma_2} I'^{(\mathcal{J}'_2, \mathcal{J}'_3)}_{d-1}$ on the left is given by $\mathcal{J}_1$ and on the right is given by $\mathcal{J}'_3$.

Similarly, a topological $(\Gamma_2, \Gamma_3)$-symmetric interface $I'^{(\mathcal{J}'_2, \mathcal{J}'_3)}_{d-1}$ from $\mathfrak{T}_{\mathcal{S}_2}^{(2)}$ to $\mathfrak{T}_{\mathcal{S}_3}^{(3)}$ can act from the right on a general (topological or non-topological) $(\Gamma_1, \Gamma_2)$-symmetric interface $I_{d-1}^{(\mathcal{J}_1, \mathcal{J}_2)}$ from $\mathfrak{T}_{\mathcal{S}_1}^{(1)}$ to $\mathfrak{T}_{\mathcal{S}_2}^{(2)}$ to give rise to a $(\Gamma_1, \Gamma_3)$-symmetric interface

$$I_{d-1}^{(\mathcal{J}_1, \mathcal{J}_2)} \otimes_{\Gamma_2} I'^{(\mathcal{J}'_2, \mathcal{J}'_3)}_{d-1}, \tag{62}$$

from $\mathfrak{T}_{\mathcal{S}_1}^{(1)}$ to $\mathfrak{T}_{\mathcal{S}_3}^{(3)}$.

**Fusion of topological interfaces after gauging.** Let $I_{d-1}^{(\mathcal{J}_1, \mathcal{J}_2)}$ and $I'^{(\mathcal{J}'_2, \mathcal{J}'_3)}_{d-1}$ encountered above be topological interfaces. After gauging, we obtain a topological interface

$$I_{d-1}^{(\mathcal{J}_1, \mathcal{J}_2)} / (\Gamma_1, \Gamma_2), \tag{63}$$

from $\mathfrak{T}_{\mathcal{S}_1}^{(1)} / \Gamma_1$ to $\mathfrak{T}_{\mathcal{S}_2}^{(2)} / \Gamma_2$, and a topological interface

$$I_{d-1}^{(\mathcal{J}_2, \mathcal{J}_3)} / (\Gamma_2, \Gamma_3), \tag{64}$$

from $\mathfrak{T}_{\mathcal{S}_2}^{(2)} / \Gamma_2$ to $\mathfrak{T}_{\mathcal{S}_3}^{(3)} / \Gamma_3$.

Their fusion is given by

$$\frac{I_{d-1}^{(\mathcal{J}_1, \mathcal{J}_2)}}{(\Gamma_1, \Gamma_2)} \otimes \frac{I_{d-1}^{(\mathcal{J}_2, \mathcal{J}_3)}}{(\Gamma_2, \Gamma_3)} = \frac{I_{d-1}^{(\mathcal{J}_1, \mathcal{J}_2)} \otimes_{\Gamma_2} I'^{(\mathcal{J}'_2, \mathcal{J}'_3)}_{d-1}}{(\Gamma_1, \Gamma_3)}, \tag{65}$$

where the right hand side is the topological interface from $\mathfrak{T}_{\mathcal{S}_1}^{(1)} / \Gamma_1$ to $\mathfrak{T}_{\mathcal{S}_3}^{(3)} / \Gamma_3$ obtained by gauging on both sides of the $(\Gamma_1, \Gamma_3)$-symmetric interface (62).

**Symmetries from topological interfaces.** Consider that we are provided a topological interface $I_{d-1}$ from a $d$-dimensional QFT $\mathfrak{T}$ to the $d$-dimensional QFT $\mathfrak{T}_{\mathcal{S}} / \Gamma$ obtained from $\mathfrak{T}$ by gauging a $\Gamma$ higher-group symmetry with coupling $\mathcal{S}$. Consider now a topological codimension-1 defect $D_{d-1}$ of $\mathfrak{T}$ along with a coupling $\mathcal{J}_L$ converting it into a topological interface from $\Gamma$-symmetric QFT $\mathfrak{T}_{\mathcal{S}}$ to the QFT $\mathfrak{T}$. This provides an interface

$$D_{d-1}^{(\mathcal{J}_L)} / \Gamma, \tag{66}$$

from QFT $\mathfrak{T}_{\mathcal{S}}/\Gamma$ to the QFT $\mathfrak{T}$, by gauging $\Gamma$ on the left of $D_{d-1}^{(\mathcal{J}_L)}$. We can now construct a topological defect of $\mathfrak{T}$ by composing the two interfaces discussed above

$$I_{d-1} \otimes \frac{D_{d-1}^{(\mathcal{J}_L)}}{\Gamma} . \tag{67}$$

Thus, we have converted the information about topological interface $I_{d-1}$ into symmetries generated by $I_{d-1} \otimes \frac{D_{d-1}^{(\mathcal{J}_L)}}{\Gamma}$ for various choices of $D_{d-1}$ and $\mathcal{J}_L$.

**Example: Non-invertible symmetries from ABJ anomalies.** [6] used this method to construct non-invertible codimension-1 topological defects in 4d gauge theories with ABJ anomalies. A continuous symmetry afflicted with an ABJ anomaly acts on a gauge theory by shifting the theta angle. Thus a topological codimension-1 defect implementing such an anomalous symmetry transformation provides an invertible interface $I_3$, or in other words a duality, between gauge theories with different values of theta angles.

Let us illustrate using one of the simplest examples appearing in [6]. Let $\mathfrak{T}^{(1)}$ be 4d $U(1)$ gauge theory with $\theta = 0$ and $\mathfrak{T}^{(2)}$ be 4d $U(1)$ gauge theory with $\theta = \pi$. Assume $\mathfrak{T}^{(1)}$ has a $U(1)$ global symmetry with an ABJ anomaly, providing an interface $I_3$ from $\mathfrak{T}^{(1)}$ to $\mathfrak{T}^{(2)}$. As discussed in [6], the theory $\mathfrak{T}^{(2)}$ can also be obtained from $\mathfrak{T}^{(1)}$ by gauging a $\mathbb{Z}_2^{(1)}$ subgroup of the magnetic $U(1)^{(1)}$ 1-form symmetry of $\mathfrak{T}^{(1)}$ along with a discrete torsion specified by the 4d $\mathbb{Z}_2^{(1)}$ SPT phase with effective action[8]

$$\frac{\mathcal{P}(B_2)}{2} . \tag{68}$$

We can construct another topological interface from $\mathfrak{T}^{(2)}$ to $\mathfrak{T}^{(1)}$ as follows. Take $\mathsf{T}$ to be a 3d TQFT with $\mathbb{Z}_2^{(1)}$ 1-form symmetry with anomaly (68) and chiral central charge a multiple of one half.[9] An example of such a theory is the semion model, or $U(1)_2$ Chern-Simons theory $\mathsf{T}_\mathsf{S}$ discussed earlier, which was also used by [6]. Stack $\mathsf{T}_\mathsf{S}$ on top of $\mathfrak{T}^{(1)}$ to obtain a codimension-1 defect $D_3^{(\mathsf{T}_\mathsf{S})}$ of $\mathfrak{T}^{(1)}$. A consequence of the anomaly (68) is that a gauge-invariant coupling $\mathcal{J}$ of $D_3^{(\mathsf{T}_\mathsf{S})}$ to $\mathbb{Z}_2^{(1)}$ backgrounds in $\mathfrak{T}^{(1)}$ is possible only if the bulk couplings to $\mathbb{Z}_2^{(1)}$ backgrounds on left and right of $D_3^{(\mathsf{T}_\mathsf{S})}$ differ by a $\mathbb{Z}_2^{(1)}$ SPT phase whose effective action is (68). Let us call the bulk coupling involving extra SPT as $\mathcal{S}'$ and the bulk coupling not involving extra SPT as $\mathcal{S}$. Then, the codimension-1 topological defect $D_3^{(\mathsf{T}_\mathsf{S})}$ of $\mathfrak{T}^{(1)}$ descends to a $\mathbb{Z}_2^{(1)}$-symmetric topological interface $D_3^{(\mathsf{T}_\mathsf{S}, \mathcal{J})}$ from the $\mathbb{Z}_2^{(1)}$-symmetric 4d QFT $\mathfrak{T}_{\mathcal{S}'}^{(1)}$ to the $\mathbb{Z}_2^{(1)}$-symmetric 4d QFT $\mathfrak{T}_{\mathcal{S}}^{(1)}$.

Gauging $\mathbb{Z}_2^{(1)}$ 1-form symmetry on both sides of $D_3^{(\mathsf{T}_\mathsf{S}, \mathcal{J})}$ we obtain a topological interface

$$I_3^{(\mathsf{S})} := D_3^{(\mathsf{T}_\mathsf{S}, \mathcal{J})}/\mathbb{Z}_2^{(1)} , \tag{69}$$

from the 4d QFT

$$\mathfrak{T}_{\mathcal{S}'}^{(1)}/\mathbb{Z}_2^{(1)} \cong \mathfrak{T}^{(2)} , \tag{70}$$

to the 4d QFT

$$\mathfrak{T}_{\mathcal{S}}^{(1)}/\mathbb{Z}_2^{(1)} \cong \mathfrak{T}^{(1)} . \tag{71}$$

Composing this topological interface with $I_3$, we obtain a topological defect

$$I_3 \otimes I_3^{(\mathsf{S})} , \tag{72}$$

of the 4d QFT $\mathfrak{T}^{(1)}$, which generates the non-invertible symmetry discussed in [6].

---

[8]Note that to make sense of this effective action, we need to restrict to spin 4d $U(1)$ gauge theories.

[9]This requirement is there to make sure that the 4d TQFT attached to $\mathsf{T}$, capturing the gravitational anomaly of $\mathsf{T}$, vanishes on spin 4-manifolds, allowing us to treat $\mathsf{T}$ as an absolute (rather than relative) spin TQFT.

## 2.5 Condensations

A final generalization of the above considerations arises by noticing that we can replace $d$-dimensional QFTs everywhere by $d$-dimensional defects of a larger $D$-dimensional QFT. The higher-group symmetries will be localized along $d$-dimensional worldvolumes of these $d$-dimensional defects, and the whole machinery (about their gauging etc.) will carry through. Such localized symmetries were discussed in detail by [5].

The above machinery then allows us to produce new $d$-dimensional defects of the $D$-dimensional QFT by gauging localized symmetries, study the fate sub-defects and sub-interfaces under such a gauging producing sub-defects and sub-interfaces of the $d$-dimensional defects obtained after localized gauging.

**Condensation defects.** The simplest example of localized symmetries is provided by the identity $d$-dimensional defect $D_d^{(\mathrm{id})}$ inside a $D$-dimensional QFT $\mathfrak{T}$. The topological defects of $\mathfrak{T}$ of dimension less than $d$ can be submerged inside the $d$-dimensional worldvolume of $D_d^{(\mathrm{id})}$ and generate the symmetries localized along $D_d^{(\mathrm{id})}$. We can then pick a higher-group symmetry $\Gamma$ among these localized symmetries and consider turning on background gauge fields for $\Gamma$ along the $d$-dimensional worldvolume occupied by $D_d^{(\mathrm{id})}$, or in other-words any $d$-dimensional subspace inside the $D$-dimensional spacetime unoccupied by any non-trivial defects of $\mathfrak{T}$.

The choice of gauge-invariant coupling $\mathcal{S}$ then allows us to define partition functions of $\mathfrak{T}$ with $\Gamma$ backgrounds localized along any $d$-dimensional subspace $M_d$ of the $D$-dimensional spacetime, such that they are invariant under background gauge transformations localized along $M_d$.

After choosing such a coupling $\mathcal{S}$, we can gauge $\Gamma$ symmetry localized along the worldvolume of $D_d^{(\mathrm{id})}$ to obtain a non-identity $d$-dimensional defect

$$D_d^{(\mathrm{id},\mathcal{S})}/\Gamma, \tag{73}$$

of the same $D$-dimensional QFT $\mathfrak{T}$, which is topological because the underlying defect $D_d^{(\mathrm{id})}$ is topological to begin with. Such topological defects $D_d^{(\mathrm{id},\mathcal{S})}/\Gamma$ were dubbed as *condensation defects* in [4].

The machinery discussed in this section then allows us to study sub-defects and sub-interfaces of condensation defects.

**Higher-categorical structure of symmetries.** In fact, we can iterate the above procedure. We can replace $D$-dimensional QFTs by $D$-dimensional defects of larger $D'$-dimensional QFTs. The machinery of this section then studies the symmetries localized along sub-defects of defects of a QFT, the gauging of such symmetries and the fate of sub-sub-defects and sub-sub-interfaces between such sub-defects.

Of course, we can keep iterating the above procedure. Turning it around, this means that we could apply all of the machinery discussed in this section not only to defects of a $d$-dimensional QFT $\mathfrak{T}$, but also to sub-defects of defects of $\mathfrak{T}$, and sub-sub-defects of sub-defects of $\mathfrak{T}$ and so on.

If we restrict ourselves to the study only of topological defects and topological sub-defects etc, then this iterative structure of defects inside defects, and their various properties is all expected to be captured in the information of a $(d-1)$-category $\mathcal{C}_\mathfrak{T}$ associated to the QFT $\mathfrak{T}$, known as the *symmetry category* of $\mathfrak{T}$. See [5,7] for more detailed discussions of the higher-categorical aspects of non-invertible symmetries.

Starting from next section, we will assume that the readers have familiarized themselves with this higher-categorical structure.

> ### Distinction between condensation and theta defects
>
> In this section, we discussed theta defects and condensation defects, both of which provide classes of non-invertible symmetries. In this box, we discuss the distinction between the two classes.
>
> First of all, the two classes are defined differently. The **theta defects** are obtained by inserting a lower $p$-dimensional TQFT $\mathsf{T}$ inside a bulk $d$-dimensional QFT $\mathfrak{T}$ and then gauging a combined symmetry $\mathcal{S}$ of $\mathsf{T}$ and $\mathfrak{T}$. On the other hand, the **condensation defects** are obtained by gauging a symmetry $\mathcal{S}$ of a $d$-dimensional QFT $\mathfrak{T}$ on a $p$-dimensional submanifold of the $d$-dimensional spacetime.
>
> However, even though the definitions are different, one may wonder whether the two classes actually turn out to be the same. This was discussed in some detail in [7], where it was concluded that the two classes are different. The reference [7] studied the two classes for $p = 2$. If $\mathcal{S}$ is a finite abelian 0-form symmetry group $\Gamma^{(0)}$, then it was shown concretely in section 4 of [7] that the two classes of defects in fact coincide. Reference [7] also provided an abstract argument that the two classes of defects coincide even if $\Gamma^{(0)}$ is a *non-Abelian* finite 0-form symmetry group. However, as pointed out in [7], if $\mathcal{S}$ is a 2-group symmetry in which 0-form and 1-form symmetries mix non-trivially, there exist theta defects that are not condensation defects, and hence the two classes of defects start to differ.
>
> For higher values of $p$, the two classes differ already for $\mathcal{S} = \Gamma^{(0)}$ as not all $\Gamma^{(0)}$ symmetric $p$-dimensional TQFTs for $p \geq 3$ admit topological boundary conditions. The existence of a $\Gamma^{(0)}$-symmetric topological boundary condition would imply that the resulting theta defect can also be constructed as a condensation defect.
>
> Finally, if we consider twisted theta defects, then they are distinct from condensation defects already for $p = 2$ and $\mathcal{S} = \Gamma^{(0)}$. Also note that the twisted theta defect (36) is not a condensation defect.

## 3 Mathematical, higher-categorical structure

In this section, we translate various special cases of the physical construction described above into well-known mathematical concepts in category theory. It should be noted that what is discussed below is only physicists' attempt at giving a mathematical definition to the physical concepts encountered in the discussion of symmetries in QFTs, but there might be various adjectives and subtleties missing from the mathematical discussion. Our purpose is to point out the relevant mathematical objects in an effort to motivate the precise mathematical treatment of the physical concepts encountered in the study of symmetries.

### 3.1 Higher vector spaces and non-anomalous topological orders

**Higher-categories of universal topological defects.** As we discussed in the previous section, one of the ways of constructing $p$-dimensional topological defects of any $d$-dimensional QFT $\mathfrak{T}$ is to simply stack a $p$-dimensional TQFT $\mathsf{T}$ on a $p$-dimensional locus inside the spacetime occupied by $\mathfrak{T}$. Thus any $d$-dimensional theory $\mathfrak{T}$ carries a *universal* sector of topological defects described by $(d-1)$-dimensional TQFTs and topological defects/interfaces of $(d-1)$-dimensional TQFTs.[10]

---

[10]Note that this automatically includes all lower dimensional TQFTs. For example, $(d-2)$-dimensional TQFTs can be viewed as topological codimension-1 defects of the completely trivial $(d-1)$-dimensional TQFT.

This universal sector is expected to be described by a monoidal $(d-1)$-category $\mathcal{T}_{d-1}$ whose objects are $(d-1)$-dimensional TQFTs and morphisms are topological defects/interfaces of $(d-1)$-dimensional TQFTs. Note that $\mathcal{T}_{d-2}$ is contained inside $\mathcal{T}_{d-1}$ as the $(d-2)$-category formed by endomorphisms (i.e. the defects) of the identity object (i.e. the trivial $(d-1)$-dimensional TQFT) of $\mathcal{T}_{d-1}$. Continuing iteratively $\mathcal{T}_p$ for every $p < d$ is contained in $\mathcal{T}_{d-1}$.

**1-category of universal line defects and vector spaces.** Let us study special cases of $\mathcal{T}_p$ higher-categories. For $p = 1$, we have

$$\mathcal{T}_1 \cong \mathsf{Vec}\,, \tag{74}$$

that is the category $\mathcal{T}_1$ of 1d TQFTs can be identified with the category $\mathsf{Vec}$ of finite-dimensional vector spaces. The identification

$$\mathcal{T}_1 \to \mathsf{Vec}\,, \tag{75}$$

is made by a bulk-boundary correspondence: A 1d TQFT $\mathsf{T} \in \mathcal{T}_1$ is mapped to the vector space $V_\mathsf{T}$ of local operators living at the 0d boundary of $\mathsf{T}$. We can also identify $V_\mathsf{T}$ with the space of states assigned to a point by $\mathsf{T}$. The fusion rule of $\mathsf{T}$ with an arbitrary line defect $D_1$ of $\mathfrak{T}$ is

$$\mathsf{T} \otimes D_1 \cong \dim(V_\mathsf{T}) D_1\,, \tag{76}$$

where $\dim(V_\mathsf{T})$ is the dimension of the vector space $V_\mathsf{T}$, and $\dim(V_\mathsf{T}) D_1$ denotes a direct sum of $\dim(V_\mathsf{T})$ copies of $D_1$.

**Universal surface defects.** For $p = 2$, we are studying 2d TQFTs. First of all, we have invertible 2d TQFTs

$$\mathsf{I}_\lambda\,, \qquad \lambda \in \mathbb{R}^+\,, \tag{77}$$

which are known in the physics literature as 'Euler number counterterms'. The partition function of such a TQFT on a 2d manifold $\Sigma_g$ of genus $g$ is

$$\lambda^{2-2g}\,. \tag{78}$$

These are the only 2d TQFTs with a single vacuum, and any 2d TQFT $\mathsf{T}$ with $n$ vacua can be decomposed as [59]

$$\mathsf{T} = \bigoplus_{i=1}^{n} \mathsf{I}_{\lambda_i}\,, \tag{79}$$

that is the TQFT $\mathsf{T}$ reduces in each vacuum to an Euler number counterterm.

**2-vector spaces.** Instead of studying 2d TQFTs, one can study 2d non-anomalous[11] topological orders, which are defined as 2d TQFTs modulo invertible 2d TQFTs [50], and form a 2-category $\mathcal{O}_2$.

There is a canonical inclusion

$$\mathcal{O}_2 \hookrightarrow \mathcal{T}_2\,, \tag{80}$$

whose image contains 2d TQFTs having $n$ vacua such that restricting to any vacuum we obtain the trivial TQFT with $\lambda = 0$. Thus $\mathcal{O}_2$ is a fusion 2-category with a single simple object (upto isomorphism) corresponding to the completely trivial 2d TQFT. Physically, this is just the well-known fact that there are no non-trivial topological orders in 2d.

We can identify

$$\mathcal{O}_2 \cong \text{2-}\mathsf{Vec}\,, \tag{81}$$

---

[11]Here anomaly refers to gravitational anomaly. The presence of this anomaly means that we are studying 2d theories that are relative, and should be properly understood as boundary conditions of 3d invertible TQFTs.

where 2-Vec is the fusion 2-category formed by '2-vector spaces', which are by definition finite[12] semi-simple abelian 1-categories. The identification

$$\mathcal{O}_2 \to \text{2-Vec}, \tag{82}$$

is made by a bulk-boundary correspondence: A 2d TQFT $\mathsf{T} \in \mathcal{O}_2 \subset \mathcal{T}_2$ is mapped to the 1-category $\mathcal{C}_\mathsf{T}$ of line operators living on the boundary of $\mathsf{T}$. If $\mathsf{T}$ has $n$ vacua, then $\mathcal{C}_\mathsf{T}$ has $n$ simple objects and can be identified as

$$\mathcal{C}_\mathsf{T} \cong n\,\text{Vec}, \tag{83}$$

where on right hand side we have a direct sum of $n$ copies of the 1-category Vec of finite dimensional vector spaces. The fusion rule of $\mathsf{T}$ with an arbitrary surface defect $D_2$ of $\mathfrak{T}$ is

$$\mathsf{T} \otimes D_2 \cong n\,D_2, \tag{84}$$

where $n\,D_2$ denotes a direct sum of $n$ copies of $D_2$.

**2-category of universal surface defects.** We can express $\mathcal{T}_2$ as

$$\mathcal{T}_2 \cong \mathcal{O}_2 \boxtimes \text{2-Vec}_{\mathbb{R}^+}, \tag{85}$$

where 2-Vec$_{\mathbb{R}^+}$ describes the Euler counter-terms, and is simply the monoidal 2-category of $\mathbb{R}^+$-graded 2-vector spaces.

**Universal 3d defects and 3-vector spaces.**[13] For $p = 3$, the 3-category $\mathcal{O}_3$ formed by 3d non-anomalous topological orders is

$$\mathcal{O}_3 \cong \text{3-Vec}, \tag{86}$$

where the right hand side is the fusion 3-category formed by '3-vector spaces', which is by definition the 3-category formed by multi-fusion 1-categories.

The identification

$$\text{3-Vec} \to \mathcal{O}_3, \tag{87}$$

is made again by bulk-boundary correspondence with the map essentially taking topological boundary conditions of a 3d TQFT to the 3d TQFT. A non-anomalous 3d TQFT $\mathsf{T}$ (upto stacking with invertible $E_8$ phases) admits topological boundary conditions.[14] Pick a topological boundary condition $\mathfrak{B}_\mathsf{T}$ of $\mathsf{T}$. $\mathfrak{B}_\mathsf{T}$ can be characterized by the topological line defects living on it. These topological line defects form a multi-fusion 1-category $\mathcal{C}_{\mathfrak{B}_\mathsf{T}}$ and the above map is simply

$$\mathcal{C}_{\mathfrak{B}_\mathsf{T}} \mapsto \mathsf{T}. \tag{88}$$

Miraculously, the category of boundary lines $\mathcal{C}_{\mathfrak{B}_\mathsf{T}}$ completely determines the TQFT $\mathsf{T}$ via the well-known Turaev-Viro construction based on $\mathcal{C}_{\mathfrak{B}_\mathsf{T}}$. In particular, the modular multi-tensor category $\mathcal{M}_\mathsf{T}$ formed by topological line defects of $\mathsf{T}$ is recovered as

$$\mathcal{M}_\mathfrak{T} \cong \mathcal{Z}\left(\mathcal{C}_{\mathfrak{B}_\mathsf{T}}\right), \tag{89}$$

---

[12]All categories we discuss are $\mathbb{C}$-linear unless otherwise stated. Note that the categories $\mathcal{C}_q^\Gamma$ discussed later are non-linear.

[13]We thank Thibault Décoppet and David Jordan for discussions regarding various points appearing in this subsection from this point onward.

[14]Such a boundary condition does not exist when we are dealing with anomalous 3d TQFTs (calling such systems as TQFTs is a misnomer, as such 3d theories are topological boundary conditions of 4d TQFTs rather than properly defined 3d theories).

where $\mathcal{Z}(\mathcal{C}_{\mathfrak{B}_\mathsf{T}})$ denotes the Drinfeld center of $\mathcal{C}_{\mathfrak{B}_\mathsf{T}}$. A different topological boundary condition $\mathfrak{B}'_\mathsf{T}$ of $\mathsf{T}$ carries a different multi-fusion category $\mathcal{C}_{\mathfrak{B}'_\mathsf{T}}$ but Turaev-Viro construction based on it produces the same 3d TQFT $\mathsf{T}$. At the level of topological line defects of $\mathsf{T}$, we have an identification

$$\mathcal{Z}\left(\mathcal{C}_{\mathfrak{B}_\mathsf{T}}\right) \cong \mathcal{Z}\left(\mathcal{C}_{\mathfrak{B}'_\mathsf{T}}\right), \tag{90}$$

of Drinfeld centers.

**Simple objects of** 3-Vec. Note that, unlike Vec and 2-Vec, the 3-category 3-Vec has an infinite number of isomorphism classes of simple objects. The isomorphism classes of simple objects of 3-Vec are characterized by isomorphism classes of modular tensor categories that can be expressed as Drinfeld centers. Equivalently, the isomorphism classes of simple objects of 3-Vec are characterized by the Morita equivalence[15] classes of fusion categories.

**More universal 3d defects for spin-QFTs.** One might wonder why we do not discuss 3d TQFTs with topological line defects described by other modular multi-tensor categories that cannot be expressed as Drinfeld centers. The reason is that such 3d TQFTs are anomalous. However, if we include extra structure, then some of these anomalous TQFTs become non-anomalous. We encountered such cases in previous section, where we saw that if $\mathfrak{T}$ is a spin QFT, then we can include 3d TQFTs with chiral central charge $c_-$ being a multiple of half. In particular if $c_- \neq 0$, then the corresponding modular tensor category $\mathcal{M}_\mathsf{T}$ cannot be expressed as a Drinfeld center and so is not included in 3-Vec.

**Is** 3-Vec **the simplest fusion 3-category?** The final issue that we would like to discuss is the following puzzle: 3-Vec is often referred to as the simplest fusion 3-category. However, as we have seen, this includes many non-trivial 3d topological orders, e.g. topological orders characterized by non-trivial modular tensor categories (albeit only those that can be expressed as Drinfeld centers). Physically, extending the 1d and 2d examples discussed above, it seems that we could study a closed set of 3d topological orders which is simpler than the above set. This simpler set comprises of the trivial 3d TQFT and its direct sums. A 3d TQFT in this simpler set has $n$ vacua such that in each vacuum the TQFT reduces to the trivial theory. Clearly, such 3d TQFTs form a rather simple monoidal 3-category that we refer to as

$$\text{3-Vec}^0, \tag{91}$$

where the subscript 0 stands for trivial, as this 3-category. In fact, just like Vec and 2-Vec, the 3-category 3-Vec$^0$ has a single simple object upto isomorphism corresponding to the trivial 3d TQFT.

The crucial part of the definition of a fusion 3-category [50] violated by the monoidal 3-category 3-Vec$^0$ is that a fusion 3-category $\mathcal{C}$ has to be Karoubi complete, which physically means that any 3d topological defect obtained by gauging a (possibly non-invertible) symmetry localized along the worldvolume of a 3d topological defect corresponding to an object of $\mathcal{C}$ should also correspond to an object of $\mathcal{C}$ [49].

This condition clearly fails for 3-Vec$^0$. For example, consider gauging the $\mathbb{Z}_2$ 0-form symmetry of the trivial 3d TQFT. This produces the 3d $\mathbb{Z}_2$ Dijkgraaf-Witten gauge theory without twist, also known as the toric code, which has a single vacuum but carries a modular tensor category of lines containing more than one simple object. This makes it clear that toric code lies outside 3-Vec$^0$.

---

[15]Recall that Morita equivalence of fusion categories $\mathcal{C}_1$ and $\mathcal{C}_2$ is equivalent to the statement that their Drinfeld centers are same $\mathcal{Z}(\mathcal{C}_1) \cong \mathcal{Z}(\mathcal{C}_2)$.

In fact, upon Karoubi completing 3-Vec$^0$, that is upon adding objects corresponding to 3d topological orders that can be produced by gauging 3d TQFTs contained in 3-Vec$^0$, we land on a fusion 3-category equivalent to 3-Vec, i.e.[16]

$$\text{Kar}(3\text{-Vec}^0) \cong 3\text{-Vec}. \tag{92}$$

We refer to 3-Vec$^0$ as a *pre-fusion 3-category*.[17]

On the other hand, Vec and 2-Vec are Karoubi complete. For example, 2d $\mathbb{Z}_2$ gauge theory has two vacua and, upto an Euler counterterm, can be identified with the element

$$2 \times \text{Vec} \in 2\text{-Vec}, \tag{93}$$

obtained as the direct sum of two copies of Vec.

**Physically motivated definition of p-Vec.** Going along the above lines, we would like to define p-Vec as the simplest fusion $p$-category. Physically, we would want p-Vec to describe the simplest $p$-dimensional non-anomalous topological orders that are closed under condensations.

For this purpose, let us begin by defining a monoidal $p$-category

$$\text{p-Vec}^0, \tag{94}$$

as the category describing the trivial $p$-dimensional TQFT and its direct sums. This category has a single simple object (upto isomorphism) corresponding to the identity codimension-1 defect.

Recall that, for a monoidal $p$-category $\mathcal{C}$, $\Omega(\mathcal{C}) := \text{End}_1(\mathcal{C})$ is a monoidal $(p-1)$-category obtained by restricting to endomorphisms of the identity object of $\mathcal{C}$. We denote by $\Omega^p(\mathcal{C})$ the monoidal category obtained by applying $\Omega^{p-1}$ to the monoidal category $\Omega(\mathcal{C})$.

We then have

$$\Omega\left(\text{p-Vec}^0\right) = (\text{p}-1)\text{-Vec}^0. \tag{95}$$

Thus, p-Vec$^0$ has a single simple 1-endomorphism (upto isomorphism) of the identity object corresponding to the identity codimension-2 defect, and so on because

$$\Omega^q\left(\text{p-Vec}^0\right) = (\text{p}-\text{q})\text{-Vec}^0. \tag{96}$$

Note that p-Vec$^0$ is always a pre-fusion category, but may or may not be a fusion category. For low-dimensional cases we have

$$\text{Vec}^0 \cong \text{Vec}, \qquad 2\text{-Vec}^0 \cong 2\text{-Vec}, \tag{97}$$

and so it is fusion, but as we saw above 3-Vec$^0$ is not fusion.

Now we can simply add condensations to p-Vec$^0$, or in other words Karoubi complete it, to obtain our desired fusion category, leading to the definition

$$\text{p-Vec} := \text{Kar}\left(\text{p-Vec}^0\right). \tag{98}$$

That is, p-Vec is the fusion category of non-anomalous $p$-dimensional topological orders admitting topological/gapped boundaries.

---

[16]To see this, first of all note that 3-Vec comprises of all topological orders admitting topological/gapped boundaries. On the other hand, 3-Vec$^0$ comprises only of (direct sums of) trivial topological order. Now, a topological order $O_1$ is obtained by (generalized) gauging another topological order $O_2$ if and only if there exists a topological interface from $O_1$ to $O_2$. If $O_2$ is trivial, then such a topological interface is the same as having a topological boundary for $O_1$. Combining these statements together, we see that Kar(3-Vec$^0$) $\cong$ 3-Vec.

[17]That is, we define a pre-fusion $p$-category $\mathcal{C}$ to be a category satisfying all the nice properties required to be a fusion category except Karoubi completion. The Karoubi completion Kar($\mathcal{C}$) of a pre-fusion category $\mathcal{C}$ is a fusion $p$-category.

**Universal $p$-dimensional defects.**   With the above definition of p-Vec, in general we have inclusions

$$\text{p-Vec}^0 \subseteq \text{p-Vec} \subseteq \mathcal{O}_p \,, \tag{99}$$

with p-Vec$^0$ being pre-fusion, and p-Vec and $\mathcal{O}_p$ being fusion $p$-categories.

## 3.2   Higher representations

**Categories associated to higher-groups.**   A $p$-group $\Gamma$ can be converted into a monoidal $q$-category $\mathcal{C}_q^\Gamma$ for $q \geq p-1$. These categories capture the topological properties of the classifying space of the $p$-group $\Gamma$. Note that the categories $\mathcal{C}_q^\Gamma$ are non-$\mathbb{C}$-linear.

Let us discuss some simple cases: To a 1-group (i.e. an ordinary group, or a 0-form symmetry group) $\Gamma = \Gamma^{(0)}$, we can first of all associate a 0-category

$$\mathcal{C}_0^{\Gamma^{(0)}} = \Gamma^{(0)} \,, \tag{100}$$

which is the group itself. The $p$-category $\mathcal{C}_p^{\Gamma^{(0)}}$ associated to $\Gamma^{(0)}$ has isomorphism classes of simple objects labeled by elements of $\Gamma^{(0)}$, whose fusion is controlled by group multiplication. The endomorphism $(p-1)$-category of any simple object is $\mathcal{C}_{p-1}^{\Gamma^{(0)}=\{\text{id}\}}$, i.e. the $(p-1)$-category associated to a trivial $\Gamma^{(0)}$.

Consider a $(p+1)$-group with only non-trivial component being a $p$-form symmetry group $\Gamma^{(p)}$. We label the corresponding monoidal categories as $\mathcal{C}_q^{\Gamma^{(p)}}$ for $q \geq p$. The $q$-category $\mathcal{C}_q^{\Gamma^{(p)}}$ has a single simple object (upto isomorphism), $\Omega^n(\mathcal{C}_p^{\Gamma^{(p)}})$ has a single simple object (upto isomorphism) for $n < p$, and $\Omega^p(\mathcal{C}_q^{\Gamma^{(p)}}) = \mathcal{C}_{q-p}^{\Gamma^{(0)}=\Gamma^{(p)}}$ where we have used the categories $\mathcal{C}_*^{\Gamma^{(0)}}$ defined above.

Consider now a 2-group $\Gamma$, which contains a 0-form group $\Gamma^{(0)}$, a 1-form group $\Gamma^{(1)}$ and a Postnikov class valued in the group cohomology[18]

$$[\omega] \in H^3\left(\Gamma^{(0)}, \Gamma^{(1)}\right) \,. \tag{101}$$

The associated 1-category $\mathcal{C}_1^\Gamma$ has simple objects labeled by elements of $\Gamma^{(0)}$ with their fusion controlled by group multiplication of $\Gamma^{(0)}$, and morphisms of the identity object labeled by elements of $\Gamma^{(1)}$ with their fusion controlled by group multiplication of $\Gamma^{(1)}$. The information of the Postnikov class is captured in the associator of simple objects of $\mathcal{C}_1^\Gamma$.

**Higher-group graded higher vector spaces.**   We can linearize the $p$-category $\mathcal{C}_p^\Gamma$ by allowing the $p$-morphisms to be valued in $\mathbb{C}$. Let us call the resulting pre-fusion $p$-category as

$$\text{p-Vec}_\Gamma^0 \,, \tag{102}$$

which can be understood as $\Gamma$-graded version of the pre-fusion category p-Vec$^0$ discussed earlier.

We can now Karoubi complete to define

$$\text{p-Vec}_\Gamma := \text{Kar}\left(\text{p-Vec}_\Gamma^0\right) \,, \tag{103}$$

which we refer to as the fusion $p$-category of $\Gamma$-graded $p$-vector spaces.

---

[18]Note that we could also have an action $\rho$ of $\Gamma^{(0)}$ on $\Gamma^{(1)}$, in which case the Postnikov class is valued in the twisted group cohomology $H_\rho^3(\Gamma^{(0)}, \Gamma^{(1)})$. We are choosing the action $\rho$ to be trivial for simplicity.

**Making a QFT higher-group symmetric.** A $d$-dimensional QFT $\mathfrak{T}$ is converted into a $\Gamma$-symmetric $d$-dimensional QFT $\mathfrak{T}_\mathcal{S}$ for a higher-group $\Gamma$ by choosing a monoidal functor

$$\mathcal{S}: \ \mathcal{C}_{d-1}^\Gamma \to \mathcal{C}_\mathfrak{T}\,, \tag{104}$$

where the target category $\mathcal{C}_\mathfrak{T}$ is the $(d-1)$-category capturing all topological defects of $\mathfrak{T}$, known as the *symmetry category*[19] of $\mathfrak{T}$. The functor $\mathcal{S}$ is what we referred to as the 'gauge-invariant *coupling* of $\mathfrak{T}$ to $\Gamma$ background gauge fields' in the previous section.

**Higher-categories of universal $\Gamma$-symmetric topological defects.** In the same way as $p$-dimensional TQFTs provide topological defects for any $d$-dimensional QFT $\mathfrak{T}$, $\Gamma$-symmetric $p$-dimensional TQFTs provide $\Gamma$-symmetric topological defects for any $\Gamma$-symmetric $d$-dimensional QFT $\mathfrak{T}_\mathcal{S}$.

This universal sector of $\Gamma$-symmetric topological defects is expected to be described by a monoidal $(d-1)$-category $\mathcal{T}_{d-1}^\Gamma$ whose objects are $\Gamma$-symmetric $(d-1)$-dimensional TQFTs and morphisms are $\Gamma$-symmetric topological defects/interfaces of $\Gamma$-symmetric $(d-1)$-dimensional TQFTs. Note that $\mathcal{T}_{d-2}^\Gamma$ is contained inside $\mathcal{T}_{d-1}^\Gamma$ as the $(d-2)$-category formed by endomorphisms (i.e. the $\Gamma$-symmetric topological defects) of the identity object (i.e. the trivial $\Gamma$-symmetric $(d-1)$-dimensional TQFT) of $\mathcal{T}_{d-1}^\Gamma$. Continuing iteratively $\mathcal{T}_p^\Gamma$ for every $p < d$ is contained in $\mathcal{T}_{d-1}^\Gamma$.

We can recognize $\mathcal{T}_p^\Gamma$ as the monoidal $p$-category of functors

$$B\mathcal{S}: \ B\mathcal{C}_{p-1}^\Gamma \to \mathcal{T}_p\,. \tag{105}$$

Here $B\mathcal{C}$ is a $p$-category built from a monoidal $(p-1)$-category $\mathcal{C}$ as follows: $B\mathcal{C}$ contains a single object and the $q$-morphisms of $B\mathcal{C}$ are $(q-1)$-morphisms of $\mathcal{C}$ (with 0-morphisms being objects).

This is easy to see: From our previous definition, a $p$-dimensional TQFT described by an object $\mathsf{T} \in \mathcal{T}_p$ is made $\Gamma$-symmetric by choosing a monoidal functor

$$\mathcal{S}: \quad \mathcal{C}_{p-1}^\Gamma \to \mathrm{End}_\mathsf{T}(\mathcal{T}_p)\,, \tag{106}$$

where $\mathrm{End}_\mathsf{T}(\mathcal{T}_p)$ is the monoidal $(p-1)$-category formed by endomorphisms of the object $\mathsf{T}$ of $\mathcal{T}_p$. But such a functor is equivalent to a functor of the form (105).

We can similarly define monoidal $p$-category $\mathcal{O}_p^\Gamma$ of $\Gamma$-protected non-anomalous $p$-dimensional topological orders as the monoidal $p$-category of functors

$$B\mathcal{S}: \ B\mathcal{C}_{p-1}^\Gamma \to \mathcal{O}_p\,. \tag{107}$$

**Higher-representations of higher-groups.** Recall that a finite-dimensional representation $V_\mathcal{S}$ of a group $\Gamma^{(0)}$ is a homomorphism

$$\mathcal{S}: \ \Gamma^{(0)} \to \mathrm{End}(V)\,, \tag{108}$$

where $V$ is a finite-dimensional vector space and $\mathrm{End}(V)$ is the space of linear maps from $V$ to itself. Such a map is equivalent to a functor

$$B\mathcal{S}: \ B\mathcal{C}_0^{\Gamma^{(0)}} \to \mathsf{Vec}\,, \tag{109}$$

---

[19]The full symmetry category may get unwieldy, so often people study the category formed by topological defects modulo invertible TQFTs, and refer to it as the symmetry category.

where Vec is the 1-category of finite-dimensional vector spaces.[20] Such monoidal functors form the category $\mathsf{Rep}(\Gamma^{(0)})$ of finite-dimensional representations of $\Gamma^{(0)}$.

We can extend the above definition by changing the target category involved in (109). Functors

$$B\mathcal{C}_0^{\Gamma^{(0)}} \to \mathcal{M}, \tag{110}$$

with $\mathcal{M}$ being a monoidal category, can be referred to as representations of $\Gamma^{(0)}$ valued in $\mathcal{M}$ and generate a monoidal category $\mathsf{Rep}_{\mathcal{M}}(\Gamma^{(0)})$.

Now we can perform the higher-categorical generalization. Functors

$$B\mathcal{C}_{p-1}^{\Gamma} \to \mathcal{M}_p, \tag{111}$$

where $\Gamma$ is a $q$-group and $\mathcal{M}_p$ is a monoidal $p$-category, are known as $p$-representations of the $q$-group $\Gamma$ valued in $\mathcal{M}_p$, and form a monoidal $p$-category $\mathsf{p\text{-}Rep}_{\mathcal{M}_p}(\Gamma)$.

Thus, $p$-dimensional $\Gamma$-symmetric universal defects (or $\Gamma$-symmetric TQFTs) form the category

$$\mathcal{T}_p^{\Gamma} \cong \mathsf{p\text{-}Rep}_{\mathcal{T}_p}(\Gamma), \tag{112}$$

and the $p$-dimensional $\Gamma$-symmetric non-anomalous topological orders form the category

$$\mathcal{O}_p^{\Gamma} \cong \mathsf{p\text{-}Rep}_{\mathcal{O}_p}(\Gamma). \tag{113}$$

We have inclusions

$$\mathsf{p\text{-}Rep}_{\mathsf{p\text{-}Vec}^0}(\Gamma) \subseteq \mathsf{p\text{-}Rep}_{\mathsf{p\text{-}Vec}}(\Gamma) \subseteq \mathsf{p\text{-}Rep}_{\mathcal{O}_p}(\Gamma), \tag{114}$$

describing $\Gamma$-symmetric topological orders with increasing levels of complexity. For example, if $\Gamma = \Gamma^{(0)}$ a 0-form symmetry group, generalizing the arguments of [7], we expect $\mathsf{p\text{-}Rep}_{\mathsf{p\text{-}Vec}^0}(\Gamma^{(0)})$ to capture $p$-dimensional $\Gamma^{(0)}$-protected topological orders in which part of $\Gamma^{(0)}$ symmetry is spontaneously broken leading to multiple vacua permuted by the $\Gamma^{(0)}$ action, such that in each vacuum a subgroup $\Gamma^{(0)'}$ of $\Gamma^{(0)}$ is spontaneously preserved and that vacuum carries additionally a $p$-dimensional SPT phase protected by $\Gamma^{(0)'}$ 0-form symmetry. In this case, the other two $p$-categories $\mathsf{p\text{-}Rep}_{\mathsf{p\text{-}Vec}}(\Gamma^{(0)})$ and $\mathsf{p\text{-}Rep}_{\mathcal{O}_p}(\Gamma^{(0)})$ capture more general $\Gamma^{(0)}$-protected topological orders including SET phases.

Let us also define

$$\mathsf{p\text{-}Rep}(\Gamma) := \mathsf{p\text{-}Rep}_{\mathsf{p\text{-}Vec}}(\Gamma), \qquad \mathsf{p\text{-}Rep}^0(\Gamma) := \mathsf{p\text{-}Rep}_{\mathsf{p\text{-}Vec}^0}(\Gamma). \tag{115}$$

**Universal theta symmetries.** Any $d$-dimensional QFT $\mathfrak{T}/\Gamma$ arising by gauging a non-anomalous higher-group $\Gamma$ symmetry of a $d$-dimensional QFT $\mathfrak{T}$ carries a universal sector (non-symmetric) topological defects descending from the universal sector of $\Gamma$-symmetric topological defects of $\mathfrak{T}$. This universal sector of topological defects is what we defined to be *theta symmetries* in the previous section. Thus, from the analysis of this section we learn that theta symmetries of $\mathfrak{T}/\Gamma$ form the monoidal $(d-1)$-category $(\mathsf{d}-1)\text{-}\mathsf{Rep}_{\mathcal{T}_{d-1}}(\Gamma)$ which can be projected down to the monoidal $(d-1)$-category $(\mathsf{d}-1)\text{-}\mathsf{Rep}_{\mathcal{O}_{d-1}}(\Gamma)$.

---

[20]This is easy to see: the vector space $V$ in (108) is the image of the single object of $B\mathcal{C}_0^{\Gamma^{(0)}}$ under $B\mathcal{S}$. The endomorphisms of $B\mathcal{C}_0^{\Gamma^{(0)}}$ form the group $\Gamma^{(0)}$ under composition, and are mapped to endomorphisms of $V$ satisfying $\Gamma^{(0)}$ group law.

**SPT phases.** We can define a $p$-dimensional $\Gamma$-protected SPT phase to be a simple object of $\mathcal{T}_p^\Gamma \cong \mathsf{p\text{-}Rep}_{\mathcal{T}_p}(\Gamma)$ whose underlying $p$-dimensional TQFT is trivial i.e. corresponds to the identity object of $\mathcal{T}_p$. In other words, a $p$-dimensional $\Gamma$-protected SPT phase is a monoidal functor

$$\mathcal{S}: \mathcal{C}_{p-1}^\Gamma \to \mathcal{T}_{p-1}. \tag{116}$$

For example, consider $\Gamma = \Gamma^{(0)}$ a 0-form symmetry and $p = 2$. Then we have $\mathcal{T}_1 \cong \mathsf{Vec}$ and so the SPT phases are monoidal functors

$$\mathcal{S}: \mathcal{C}_1^{\Gamma^{(0)}} \to \mathsf{Vec}, \tag{117}$$

which are known to be classified (upto isomorphism) by the group cohomology $H^2(\Gamma^{(0)}, U(1))$ recovering the well-known classification of such SPT phases.

**Projective higher-representations.** Just like we defined higher-representations above, we can define projective higher-representations of a $q$-group $\Gamma$ as follows. To define projective $p$-representations, we need to first of all choose a target monoidal $(p+1)$-category $\mathcal{M}_{p+1}$ and an object $\mathsf{I}_{p+1} \in (\mathsf{p}+1)\text{-}\mathsf{Rep}_{\mathcal{M}_{p+1}}(\Gamma)$ such that $\mathfrak{I}_{p+1}$ is a monoidal functor of the form

$$\mathfrak{I}_{p+1}: \mathcal{C}_p^\Gamma \to \Omega\mathcal{M}_{p+1}, \tag{118}$$

or in other words, $\mathfrak{I}_{p+1}$ makes the identity object of $\mathcal{M}_{p+1}$ symmetric under $\Gamma$. Then, we define a projective $p$-representation of $\Gamma$ to be a 1-morphism from $\mathfrak{I}_{p+1}$ to the identity object of $(\mathsf{p}+1)\text{-}\mathsf{Rep}_{\mathcal{M}_{p+1}}(\Gamma)$. Such morphisms form a (non-monoidal) $p$-category which we call the $p$-category of projective $p$-representations of $\Gamma$ lying in the class $\mathfrak{I}_{p+1}$ and denote the $p$-category as

$$\mathsf{p\text{-}Rep}_{\mathcal{M}_{p+1}}^{\mathfrak{I}_{p+1}}(\Gamma). \tag{119}$$

As an example, consider $\Gamma = \Gamma^{(0)}$ a 0-form symmetry group, $p = 1$ and $\mathcal{M}_2 = 2\text{-}\mathsf{Vec}$. The possible objects $\mathfrak{I}_2$ are SPT phases classified (upto isomorphism) by elements of $[\alpha] \in H^2(\Gamma^{(0)}, U(1))$ discussed above. Let us choose an SPT phase described by a representative $\alpha$ of a class $[\alpha]$. The morphism category $1\text{-}\mathsf{Rep}_{2\text{-}\mathsf{Vec}}^\alpha(\Gamma^{(0)})$ then describes the usual projective representations of the group $\Gamma^{(0)}$ lying in the class $[\alpha]$.

**$\Gamma$-anomalous TQFTs.** A $p$-dimensional $\Gamma$-anomalous TQFT $\mathsf{T}_\mathcal{S}$, or in other words a $p$-dimensional (gravitationally non-anomalous) TQFT $\mathsf{T}$ with a coupling $\mathcal{S}$ of $\mathsf{T}$ to $\Gamma$ background fields afflicted with a 't Hooft anomaly, is defined as a projective representation

$$\mathsf{T}_\mathcal{S} \in \mathsf{p\text{-}Rep}_{\mathcal{T}_{p+1}}^{\mathfrak{I}_{p+1}}(\Gamma), \tag{120}$$

where the SPT phase $\mathfrak{I}_{p+1} \in \mathcal{T}_{p+1}^\Gamma$ is known as the $(p+1)$-dimensional anomaly theory capturing the 't Hooft anomaly associated to the $p$-dimensional $\Gamma$-anomalous TQFT $\mathsf{T}_\mathcal{S}$.

## 3.3 Modules and bimodules

In this subsection, we very roughly sketch how non-universal $\Gamma$-symmetric (topological or non-topological) defects of a $\Gamma$-symmetric $d$-dimensional QFT $\mathfrak{T}_\mathcal{S}$ can be constructed.

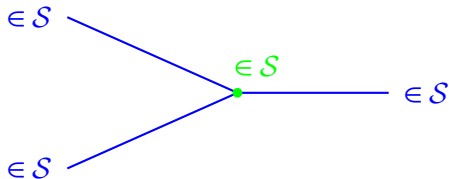

Figure 8: The choice (104) of coupling $\mathcal{S}$ is not only a choice of topological operators (shown in blue) labeled by elements of the higher-form symmetry groups $\Gamma^{(p)}$ part of the higher-group $\Gamma$, but also the choice of junctions (shown in green), and junctions of junctions etc, of such topological operators. The label '$\in \mathcal{S}$' simply indicates that the defect is in the collection $\mathcal{S}$ of defects.

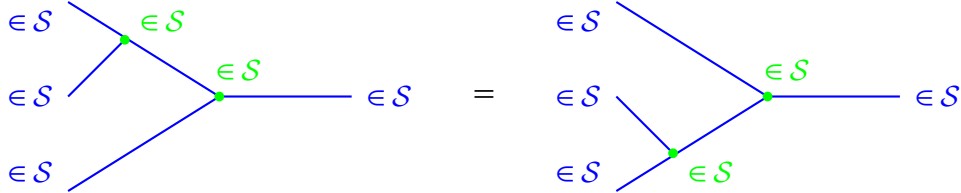

Figure 9: The figure depicts one of the possible conditions that topological operators in $\mathcal{S}$ have to satisfy for $\mathfrak{T}_{\mathcal{S}}$ to be a $\Gamma$-symmetric $d$-dimensional QFT. In fact, any two string diagrams related by a topological rearrangement of topological operators in $\mathcal{S}$ (such that the topological move leaves the boundary of the string diagram invariant) need to be equal.

**Choice of bulk coupling $\mathcal{S}$.** The choice (104) of the coupling $\mathcal{S}$ converting a $d$-dimensional QFT $\mathfrak{T}$ into a $\Gamma$-symmetric $d$-dimensional QFT $\mathfrak{T}_{\mathcal{S}}$ can be understood as a choice of topological operators in $\mathfrak{T}$ generating the $\Gamma$ symmetry along with the choice of operators at their junctions. See figure 8.

The fact that $\mathcal{S}$ is gauge-invariant means that two string diagrams composed out of operators involved in $\mathcal{S}$ and related by a topological move (that leaves the boundary of the string diagram invariant) are equal. An example is shown in figure 9.

**Choice of defect coupling $\mathcal{J}$.** Begin with a $p$-dimensional (topological or non-topological) defect $D_p$ of $\mathfrak{T}$. The coupling $\mathcal{J}$ is a choice of topological operators sitting at the junctions of $D_p$ and the bulk topological operators generating the $\Gamma$ symmetry, as shown in figure 10. The demand that $\mathcal{J}$ be a gauge-invariant coupling requires all string-diagrams (in the symmetry $(d-1)$-category $\mathcal{C}_{\mathfrak{T}}$ associated to $\mathfrak{T}$) comprising of topological operators in $\mathcal{J}$ and $\mathcal{S}$ to be the same, if they are related by topological moves (without changing the boundary of the string diagram). See figure 11.

Let us note that the above information is not sufficient to specify a codimension-1 $\Gamma$-symmetric defect. In this case the coupling $\mathcal{J}$ needs to be refined into left and right couplings $\mathcal{J}_L$ and $\mathcal{J}_R$. We discuss this refinement later in this subsection.

In favorable situations, when there are no associators (coherence relations) for $\Gamma$ topological defects in the presence of $D_p$, the choice of coupling amounts to the choice of a monoidal functor

$$\mathcal{J} : \mathcal{C}_{p-1}^{\Gamma} \to \mathcal{C}_{D_p} \, , \tag{121}$$

where $\mathcal{C}_{D_p}$ is the monoidal $(p-1)$-category describing symmetries localized along $D_p$. If $D_p$ is topological, it is an object of the $p$-category $\Omega^{d-p}(\mathcal{C}_{\mathfrak{T}})$ and we have

$$\mathcal{C}_{D_p} = \mathrm{End}_{D_p}\left(\Omega^{d-p}(\mathcal{C}_{\mathfrak{T}})\right) \, , \tag{122}$$

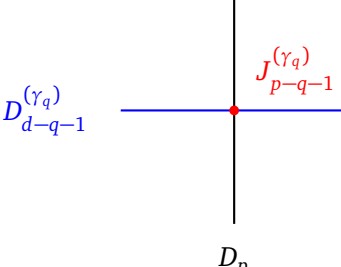

Figure 10: To define a coupling $\mathcal{J}$ of $D_p$ to $\Gamma$ background gauge fields, we need to choose topological operators lying at the junctions of $D_p$ and topological operators generating $\Gamma$ higher-group symmetry. In the figure we have illustrated such a junction operator $J_{p-q-1}^{(\gamma_q)}$ lying at the junction of $D_p$ with a bulk topological codimension-$(q+1)$ operator labeled by element $\gamma_q \in \Gamma^{(q)}$, namely the $q$-form symmetry component of $\Gamma$. This is some of the most basic data of $\mathcal{J}$. We also need to choose junctions of $D_p$ with the junctions (shown in green in figure 8) of topological operators $D_{d-q-1}^{(\gamma_q)}$, and junctions of $D_p$ with junctions of junctions of $D_{d-q-1}^{(\gamma_q)}$ etc.

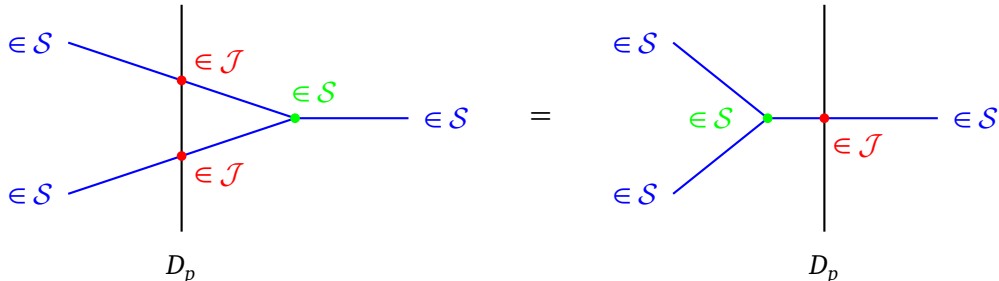

Figure 11: The figure depicts one of the possible conditions that topological operators in $\mathcal{J}$ (shown in red) have to satisfy given a set of topological operators in $\mathcal{S}$ (shown in blue and green) for $D_p^{(\mathcal{J})}$ to be a $\Gamma$-symmetric $p$-dimensional defect of $\mathfrak{T}_{\mathcal{S}}$. In fact, any two string diagrams related by a topological rearrangement of topological operators in $\mathcal{J}$ and $\mathcal{S}$ (such that the topological move leaves the boundary of the string diagram invariant) need to be equal.

is the endomorphism $(p-1)$-category of $D_p \in \Omega^{d-p}(\mathcal{C}_{\mathfrak{T}})$. The map from topological interfaces described in the previous paragraph to the topological sub-defects of $D_p$ chosen by the above functor $\mathcal{J}$ is obtained by a folding operation, see figure 12.

One can always implement the folding operation to convert information about $\mathcal{J}$ into a map of the form (121), but it will in general not be a monoidal functor. There will be obstructions for example of the type shown in figure 13.

**Choice of interface couplings $(\mathcal{J}_L, \mathcal{J}_R)$.** Similarly, we can construct $(\Gamma_L, \Gamma_R)$-symmetric interfaces from $\Gamma_L$-symmetric QFT $\mathfrak{T}_{\mathcal{S}_L}^{(L)}$ to $\Gamma_R$-symmetric QFT $\mathfrak{T}_{\mathcal{S}_R}^{(R)}$. Let $I_{d-1}$ be an interface from $\mathfrak{T}^{(L)}$ to $\mathfrak{T}^{(R)}$. A coupling $\mathcal{J}_L$ of $I_{d-1}$ to $\Gamma_L$ backgrounds is a choice of topological defects living at the ends of the bulk topological defects generating $\Gamma_L$ with coupling $\mathcal{S}_L$ along the world-volume of $I_{d-1}$. See figure 14. The coupling $\mathcal{J}_L$ is required to be left-gauge-invariant which imposes equality of two string diagrams involving topological defects in $\mathcal{S}_L$ and $\mathcal{J}_L$ related by a topological move (which leaves the boundary of the string diagram invariant). See figure 15. Similarly, we define a right-gauge-invariant coupling $\mathcal{J}_R$ of $I_{d-1}$ to $\Gamma_R$ backgrounds. Finally, for

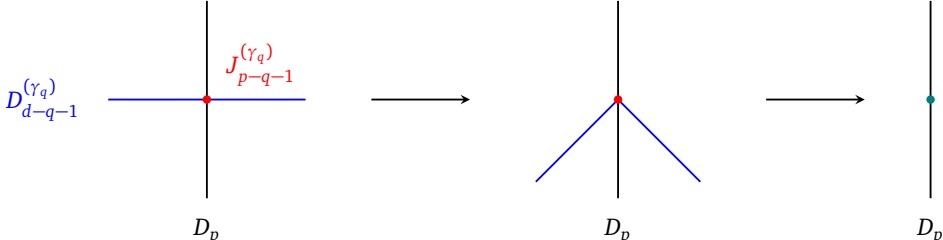

Figure 12: Folding the bulk $\Gamma$ topological operators away converts elements of $\mathcal{J}$ into topological operators localized on $D_p$. Thus, the folding operation packages the information of $\mathcal{J}$ into a map of the form (121). However, this map is a monoidal functor only if there are no non-trivial associators for $\Gamma$ topological operators in the presence of $D_p$.

the combined coupling $\mathcal{J} = (\mathcal{J}_L, \mathcal{J}_R)$ to be fully gauge-invariant, we have to impose equality of string diagrams involving topological defects in $\mathcal{S}_L, \mathcal{S}_R, \mathcal{J}_L$ and $\mathcal{J}_R$ under topological moves. See figure 16.

In fact, above we did not describe full information for converting a codimension-1 defect $D_{d-1}$ of $\mathfrak{T}$ into a $\Gamma$-symmetric codimension-1 defect $D_{d-1}^{(\mathcal{J})}$ of $\mathfrak{T}_{\mathcal{S}}$. More precisely, $D_{d-1}^{(\mathcal{J})}$ is obtained as a $(\Gamma, \Gamma)$-symmetric interface from $\mathfrak{T}_{\mathcal{S}}$ to $\mathfrak{T}_{\mathcal{S}}$, and so the coupling $\mathcal{J}$ needs to be refined into a left coupling $\mathcal{J}_L$ and a right coupling $\mathcal{J}_R$. The above topological junctions describing $\mathcal{J}$ are obtained by combining the topological ends describing $\mathcal{J}_L$ and $\mathcal{J}_R$ as shown in figure 17.

**Implementing the conditions in 2d.** The implementation of the above set of conditions is best understood for 2d QFTs and $\Gamma = \Gamma^{(0)}$ a 0-form symmetry group. This was reviewed using the modern language of symmetries in [41].

First of all, the condition shown in figure 9 means that the topological operators describing $\mathcal{S}$ form an algebra $A$ in the symmetry category $\mathcal{C}_{\mathfrak{T}}$ of the 2d QFT $\mathfrak{T}$. The algebra $A$ can be identified as the image of the canonical algebra (involving a direct sum of all simple objects) of the non-linear category $\mathcal{C}_1^{\Gamma^{(0)}}$ under the functor (104).

Coupling $\mathcal{J}_L$ shown in figure 14 satisfying the condition shown in figure 15 converts $(I_{d-1}, \mathcal{J}_L)$ into a left module for the algebra $A_L$. That is, the category of $\Gamma_L^{(0)}$-symmetric topological interfaces from a $\Gamma_L^{(0)}$-symmetric 2d QFT $\mathfrak{T}_{\mathcal{S}_L}^{(L)}$ to a 2d QFT $\mathfrak{T}^{(R)}$ is the category

$$\mathsf{Mod}_{\mathcal{M}_{L,R}}(A_L), \tag{123}$$

of $A_L$ modules in the left-module category $\mathcal{M}_{L,R}$ of the symmetry category $\mathcal{C}_{\mathfrak{T}_L}$ describing topological interfaces from $\mathfrak{T}^{(L)}$ to $\mathfrak{T}^{(R)}$.

Similarly, coupling $\mathcal{J}_R$ shown in figure 14 satisfying the condition shown in figure 15 converts $(I_{d-1}, \mathcal{J}_R)$ into a right module for the algebra $A_R$. That is, the category of $\Gamma_R^{(0)}$-symmetric topological interfaces from a 2d QFT $\mathfrak{T}^{(L)}$ to a $\Gamma_R^{(0)}$-symmetric 2d QFT $\mathfrak{T}_{\mathcal{S}_R}^{(R)}$ is the category

$$\mathsf{Mod}_{\mathcal{M}_{L,R}}(A_R), \tag{124}$$

of $A_R$ modules in the right-module category $\mathcal{M}_{L,R}$ of the symmetry category $\mathcal{C}_{\mathfrak{T}_R}$ describing topological interfaces from $\mathfrak{T}^{(L)}$ to $\mathfrak{T}^{(R)}$.

Finally, imposing also the condition shown in figure 16 converts $(I_{d-1}, \mathcal{J}_L, \mathcal{J}_R)$ into a bimodule for the algebras $(A_L, A_R)$. That is, the category of $(\Gamma_L^{(0)}, \Gamma_R^{(0)})$-symmetric topological interfaces from a $\Gamma_L^{(0)}$-symmetric 2d QFT $\mathfrak{T}_{\mathcal{S}_L}^{(L)}$ to a $\Gamma_R^{(0)}$-symmetric 2d QFT $\mathfrak{T}_{\mathcal{S}_R}^{(R)}$ is the category

$$\mathsf{Bimod}_{\mathcal{M}_{L,R}}(A_L, A_R), \tag{125}$$

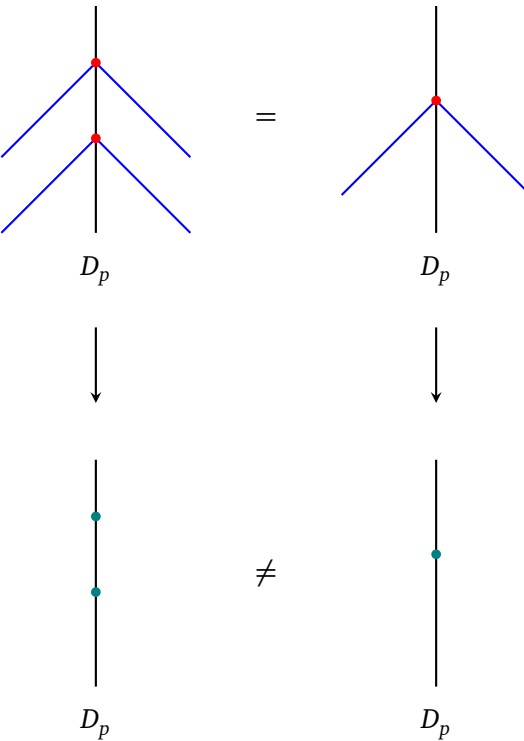

Figure 13: If there are non-trivial associators in the presence of $D_p$, then fusing and folding operations do not commute, and hence the topological operators obtained after folding (shown in teal) do not obey higher-group fusion laws.

of $(A_L, A_R)$ bimodules in the bimodule category $\mathcal{M}_{L,R}$ of the symmetry categories $(\mathcal{C}_{\mathfrak{T}_L}, \mathcal{C}_{\mathfrak{T}_R})$ describing topological interfaces from $\mathfrak{T}^{(L)}$ to $\mathfrak{T}^{(R)}$.

As a corollary, the $\Gamma^{(0)}$-symmetric topological lines of a $\Gamma^{(0)}$-symmetric 2d QFT $\mathfrak{T}_{\mathcal{S}}$ form the tensor category

$$\mathrm{Bimod}_{\mathcal{C}_{\mathfrak{T}}}(A), \tag{126}$$

of $A$-bimodules in the symmetry category $\mathcal{C}_{\mathfrak{T}}$.

**Higher-dimensions.** In a similar fashion, [8, 9] implemented the various conditions discussed in this subsection for $d = 3$ QFTs $\mathfrak{T}$ with very special choices of symmetry 2-categories $\mathcal{C}_{\mathfrak{T}}$ and special classes of 2-group symmetries. A systematic exploration of the conditions discussed here in various dimensions with various symmetry categories and for various types of higher-groups would be very interesting to tackle in future works.

# 4 A program for classification of non-invertible symmetries

In section 2, we provided a rather general physical formalism, based on gauging of invertible symmetries, that can be used in a variety of ways to construct non-invertible symmetries. In fact, we showed that several constructions of non-invertible symmetries of higher-dimensional QFTs appearing in recent literature describe special examples of the overarching structure presented here.

In section 3, we attempted to formalize the physical construction of section 2 into precise mathematical objects. We were successful at formalizing parts of the structure, while for

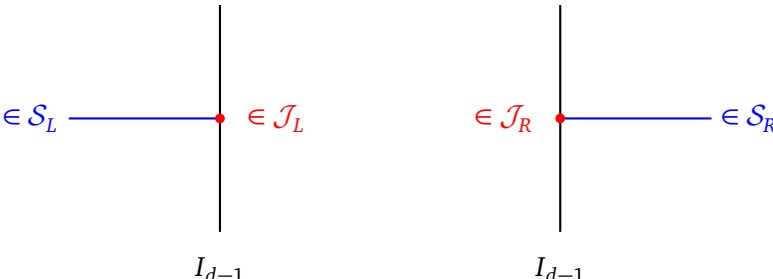

Figure 14: To define a coupling $\mathcal{J}_L$ of an interface $I_{d-1}$ to $\Gamma_L$ background gauge fields on the left, we need to choose topological operators lying at the ends along $I_{d-1}$ of topological operators generating $\Gamma_L$ higher-group symmetry. Similarly, a coupling $\mathcal{J}_R$ of $I_{d-1}$ to $\Gamma_R$ background gauge fields on the right involves choosing topological operators lying at the ends along $I_{d-1}$ of topological operators generating $\Gamma_R$ higher-group symmetry. These are some of the most basic data of $\mathcal{J}_L$ and $\mathcal{J}_R$. We also need to choose ends along $I_{d-1}$ of the junctions (shown in green in figure 8) of topological operators generating $\Gamma_L$ and $\Gamma_R$, and ends along $I_{d-1}$ of junctions of junctions of topological operators generating $\Gamma_L$ and $\Gamma_R$ etc.

the remaining parts we provided an intuitive approach in subsection 3.3 that can be made mathematically precise using the machinery of higher-category theory.

Thus, section 3 should be viewed as providing all the essential mathematical ideas required to make the physical construction of section 2 concrete and amenable to computations. Using these ideas, one should be able to concretely construct many different kinds of non-invertible symmetries carrying out the procedures detailed in section 2, as discussed below:

- First of all, one can consider understanding the **universal symmetries** that every QFT admits. This part is obtained by stacking decoupled lower-dimensional TQFTs on top of the QFT, and is characterized by the $(d-1)$-category $\mathcal{T}_{d-1}$ for a $d$-dimensional QFT $\mathfrak{T}$. Universality means that $\mathcal{T}_{d-1}$ is always a sub-category of the full symmetry $(d-1)$-category $\mathcal{C}_{\mathfrak{T}}$ of the QFT $\mathfrak{T}$ regardless of the choice of $\mathfrak{T}$. A full understanding of this universal piece is equivalent to the classification of TQFTs in various dimensions.

- At the next step, one can consider understanding another class of universal symmetries admitted by any $d$-dimensional QFT $\mathfrak{T}_{\mathcal{S}}/\Gamma$ obtainable from another $d$-dimensional QFT $\mathfrak{T}$ by gauging a non-anomalous higher-group symmetry $\Gamma$. This part is characterized by $\Gamma$-symmetric TQFTs of dimension less than $d$ and forms a $(d-1)$-subcategory $(\text{d}-1)\text{-Rep}_{\mathcal{T}_{d-1}}(\Gamma)$ of the full symmetry $(d-1)$-category $\mathcal{C}_{\mathfrak{T}}$ of $\mathfrak{T}$. We have called such symmetries as **theta symmetries** as their construction is similar to the construction of theta angle.

  Some examples of theta symmetries were concretely discussed in great computational detail in [7,8]. They analyzed the categories $2\text{-Rep}_{\mathcal{T}_1}(\Gamma) = 2\text{-Rep}_{\text{Vec}}(\Gamma) = 2\text{-Rep}(\Gamma)$ for $\Gamma$ a 2-group symmetry, which includes purely 0-form symmetry and purely 1-form symmetry. Extension to the computation of $(\text{d}-1)\text{-Rep}_{\mathcal{T}_{d-1}}(\Gamma)$ for other values of $d$ and various types of higher-groups $\Gamma$ would be an interesting problem to tackle in future works.

- The next level of complexity is the construction of **non-universal** symmetries of $\mathfrak{T}_{\mathcal{S}}/\Gamma$, namely those symmetries that arise from those topological defects of $\mathfrak{T}$ that cannot be constructed by stacking TQFTs on top of $\mathfrak{T}$.

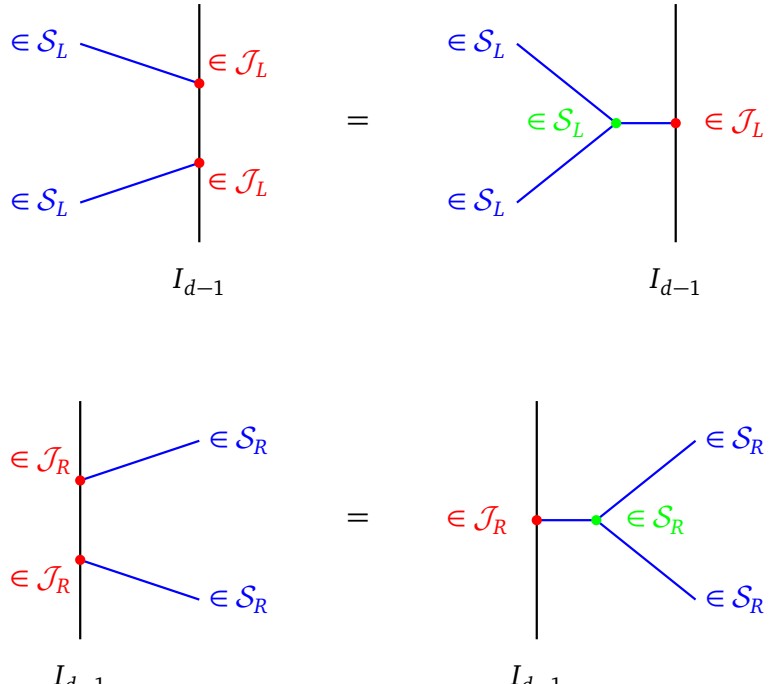

Figure 15: The top figure depicts one of the possible conditions that topological operators in $\mathcal{J}_L$ (shown in red) have to satisfy given a set of topological operators in $\mathcal{S}_L$ (shown in blue and green) for $I_{d-1}^{(\mathcal{J}_L)}$ to be a $\Gamma_L$-symmetric interface from $\mathfrak{T}_{\mathcal{S}_L}^{(L)}$ to $\mathfrak{T}^{(R)}$. In fact, any two string diagrams related by a topological rearrangement of topological operators in $\mathcal{J}_L$ and $\mathcal{S}_L$ (such that the topological move leaves the boundary of the string diagram invariant) need to be equal. The bottom figure depicts a similar condition involving $\mathcal{J}_R$ and $\mathcal{S}_R$ for $I_{d-1}^{(\mathcal{J}_R)}$ to be a $\Gamma_R$-symmetric interface from $\mathfrak{T}^{(L)}$ to $\mathfrak{T}_{\mathcal{S}_R}^{(R)}$.

A well-known example of such symmetries are provided by the duality defects of 4d QFTs discussed in [2,3]. A systematic exploration of various kinds of possible duality defects utilizing all the ingredients described in section 2 would be an interesting problem to tackle in future works.

Such symmetries are also systematically studied in the upcoming paper [9] for $\Gamma = \Gamma^{(0)}$ a 0-form symmetry group and $d = 3$. Recall that we did not provide a precise mathematical recipe for the computation of such non-universal symmetries, but rather sketched some mathematical ideas in subsection 3.3. Thus, the upcoming paper [9] should be viewed as evidence that the ideas of subsection 3.3 can be converted into precise mathematical computations that, despite the occurrence of many subtleties, can be concretely carried out. We provide numerous checks for the validity of these computations in [9] and encounter interesting phenomena like symmetry fractionalization on top of condensation defects.

Generalizing the analysis of [9] to arbitrary 2-groups $\Gamma$ in $d = 3$, and/or to higher $d$ would be a very interesting direction for future research.

- A generalization of the above non-universal symmetries involves the understanding of topological interfaces from $\mathfrak{T}_{\mathcal{S}}/\Gamma$ to $\mathfrak{T}$ starting from codimension-1 topological defects of $\mathfrak{T}$. Such interfaces can be composed with known interfaces from $\mathfrak{T}$ to $\mathfrak{T}_{\mathcal{S}}/\Gamma$ to construct new codimension-1 topological defects of $\mathfrak{T}$.

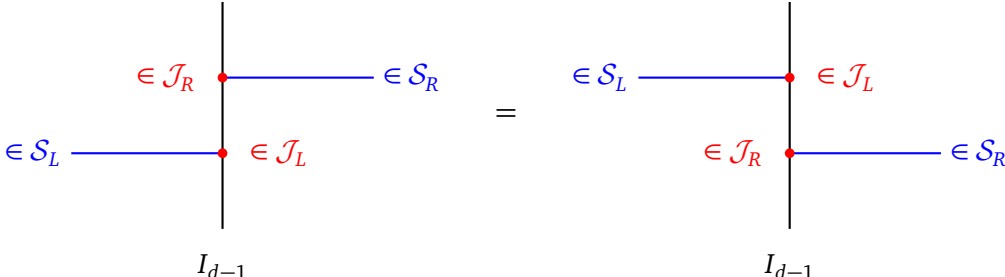

Figure 16: The figure depicts one of the possible conditions that topological operators in $\mathcal{J}_L$ and $\mathcal{J}_R$ (shown in red) have to satisfy given a set of topological operators in $\mathcal{S}_L$ and $\mathcal{S}_R$ (shown in blue) in order for $I_{d-1}^{(\mathcal{J}_L, \mathcal{J}_R)}$ to be a $(\Gamma_L, \Gamma_R)$-symmetric interface from $\mathfrak{T}_{\mathcal{S}_L}^{(L)}$ to $\mathfrak{T}_{\mathcal{S}_R}^{(R)}$. In fact, any two string diagrams related by a topological rearrangement of topological operators in $\mathcal{J}$ and $\mathcal{S}$ (such that the topological move leaves the boundary of the string diagram invariant) need to be equal.

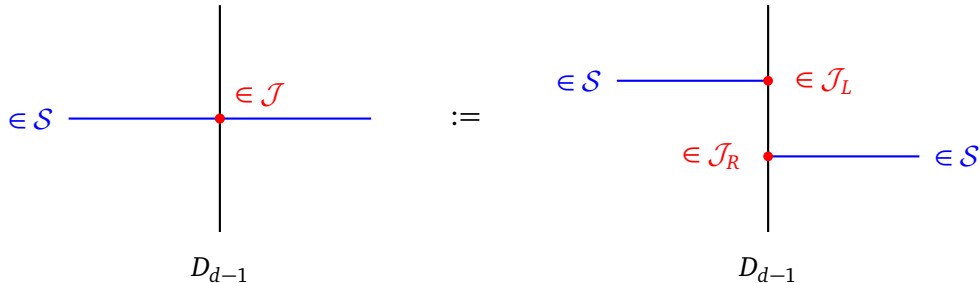

Figure 17: The topological junctions in $\mathcal{J}$ for a $\Gamma$-symmetric codimension-1 defect $D_{d-1}^{(\mathcal{J})}$ are actually a combination of the topological ends in $\mathcal{J}_L$ and $\mathcal{J}_R$ converting $D_{d-1}$ into a $(\Gamma, \Gamma)$-symmetric interface $D_{d-1}^{(\mathcal{J}_L, \mathcal{J}_R)}$ from $\mathfrak{T}_\mathcal{S}$ to $\mathfrak{T}_\mathcal{S}$.

An example of this procedure was used by [6] to construct non-invertible symmetries of 4d QFTs using ABJ anomalies. A systematic exploration of such topological interfaces has not been undertaken yet even for simple examples of $\Gamma$, $d$ and $\mathcal{C}_\mathfrak{T}$. It would be a very interesting problem to tackle in future works. It should be noted that such topological interfaces provide examples of "non-invertible dualities" between $\mathfrak{T}_\mathcal{S}/\Gamma$ and $\mathfrak{T}$, so would be interesting to explore on their own as generalizations of the standard invertible dualities.

• It should be noted that all of the above methods are also applicable to the construction of condensation defects by simply replacing QFT $\mathfrak{T}$ by the identity defect (of some dimension) in a QFT $\mathfrak{T}$ and more generally to the construction of new topological defects by gauging symmetries localized on an arbitrary topological defect in a QFT.

A systematic analysis of condensation defects was performed in [4, 7] for $d = 3$ and gauging of 1-form symmetries on surfaces. Extensions of these works to higher dimensions and gauging of other kinds of higher-group symmetries, and also extensions to gauging of symmetries localized on non-identity defects would be interesting problems to tackle in future works.

## Acknowledgments

We thank Thibault Décoppet, Clement Delcamp, David Jordan, Jingxiang Wu for discussions and Lea Bottini for collaboration on closely related work [9].

**Funding information** Part of this work has been carried out while two of the authors (LB, SSN) were at the Aspen Center for Physics, which is supported by National Science Foundation grant PHY-1607611. This work is supported by the European Union's Horizon 2020 Framework through the ERC grants 682608 (LB, SSN) and 787185 (LB). SSN acknowledges support through the Simons Foundation Collaboration on "Special Holonomy in Geometry, Analysis, and Physics", Award ID: 724073, Schäfer-Nameki. AT is supported by the Swedish Research Council (VR) through grants number 2019-04736 and 2020-00214.

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
