# Peer review of "Unifying Constructions of Non-Invertible Symmetries"

_SciPost Physics, doi:SciPost Phys. 15, 122 (2023)_

## Round 1 · Referee Report · Anonymous (Referee 1) · 2023-6-14

Weaknesses

The paper is not very readable due to the style of presentation. When the authors tried to explain a new concept, they kept introducing new notations, with more subscripts and/or more superscripts. Even though the article contains several examples, which should be a positive point of this paper and should have been illustrative for the readers, the style of writing makes them very hard to digest.

Report

This paper focuses on a method of constructing non-invertible symmetries by first considering a product between (1) a quantum field theory with a given non-anomalous global symmetry and (2) a topological quantum field theory (TQFT) with that global symmetry, and then gauging the diagonal global symmetry in the combined system of (1) and (2). The TQFT is then regarded as a topological defect, which is a symmetry generator, in the theory. Such a symmetry is referred to as a theta symmetry and the corresponding defect is called a theta defect by the authors. Suppose that the aforementioned symmetry has an 't Hooft anomaly. The defect can be "decorated" by a TQFT with the 't Hooft anomaly that cancels the former. Such a defect is referred to as a twisted theta defect in the paper. The authors then formulate the (twisted) theta defects in terms of higher-categorical structures. Such an idea of constructing non-invertible symmetries is very similar to that of constructing condensation defects, whereby a mesh of the topological defects associated to the global symmetry is inserted on a higher co-dimensional surface. This process is known as higher-gauging. The authors indeed pointed out that every condensation defect is a theta defect. A natural question that arises is whether there is any theta defect which is not a condensation defect. To put it more provocatively, why is the term "theta defects" necessary given that the terms "condensation defects" and "higher-gauging" have already existed and commonly used in the literature?

Since the concept of "theta defects" could be useful in the future and there has been a companion paper [arXiv:2212.06842] that contains many interesting examples, the referee recommends this article for publication after the points in Section "Requested changes" are addressed satisfactorily.

Requested changes

  1. The authors should really justify carefully why the term "theta defects" is necessary. Is there an example of a theta defect which is not a condensation defect? What is the advantage of the new term "theta defect" over the old term "condensation defect"?
  2. The authors wrote in the abstract about a concrete computational scheme. It would be a good idea to point out concretely and specifically which computations can be done and which results that can be obtained using theta defects.
  3. Below Eq. (6.47) of [arXiv:2204.09025], the authors discussed the domain wall decorated with a TQFT so as to make it free of gauge anomalies. Is this related to the twisted theta defect discussed in this article? If so, the authors should refer to this result.
  4. The authors mentioned intrinsic non-invertible symmetries only in passing in the introduction. How does this fit into the program for classification of non-invertible symmetries proposed in Section 4 of this paper? If this is left for future work, does this mean that the program discussed in Section 4 is an attempt to classify only non-intrinsic non-invertible symmetries?

---

## Round 1 · Referee Report · Anonymous (Referee 2) · 2023-6-26

Report

This paper provides a more unified approach to non-invertible symmetries than has appeared in the literature so far, with the basic technique being stacking of a QFT with a lower-dimensional TQFT, followed by potential gaugings. (One could also consider stacking with non-topological QFTs to obtain more general, non-topological defects). This gives rise to what the authors refer to as "theta defects." More generally, given an anomalous topological defect in a QFT, it is sometimes possible to stack with a TQFT of the same dimension and opposite anomaly to obtain a gauge-invariant defect, which then survives gauging to become a "twisted theta defect."

One of the nice aspects of this paper is that they recast many of the above concepts in a clean and streamlined mathematical language. In contrast to the first referee, I found the paper quite readable. But I do have one complaint, regarding the claim that they provide a "concrete computational scheme" for non-invertible symmetries. While I agree that this is the case for what they call "theta defects," it is far from the case for "twisted theta defects," as the authors themselves note. But twisted theta defects are really the most interesting ones---the others are basically just classified by listing lower-dimensional TQFTs and potential couplings to bulk gauge fields! This is not to detract from their results, but I personally think they would benefit from more transparent wording.

Altogether I think this is a nice, well-written paper with interesting results and would recommend it for publication in SciPost. I have only a few minor suggested changes, which I will leave as optional for the authors:

Requested changes

  1. The authors sometimes use the term "anomalous" to refer to theories that are relative, which is rather confusing (especially since they simultaneously use the word to refer to normal `t Hooft anomalies).  Personally, I would suggest avoiding this terminology altogether, but because the authors do address this issue in Footnote 14, I will leave this as an optional change.

  2. There are a couple of places in Section 3 where citations could be added. For example, please include citations for the following claims: 2a. Below (3.14): "A non-anomalous 3d TQFT $\mathsf{T}$ admits topological boundary conditions." 2b. The claim that monoidal functors (3.44) are classified by second group cohomology. It's also not obvious to me that the map in (3.35) is equivalent to a monoidal functor (3.36), so I would appreciate a citation or sentence of explanation; whichever you see fit.

  3. A couple of minor typos: 3a. In the sentence at the top of page 38, one of the "subgroup" in the phrase "subgroup $\Gamma(0)'$ subgroup " should be deleted. 3b. In (3.51), I believe that $A_L$ should be $A_R$?

  4. If I understood correctly, in Sections 2 and 4 (e.g. the bottom of page 47) you draw a distinction between "twisted theta defects" and a "generalization" thereof which comes from composing interfaces. But are these necessarily distinct? For example, you show that the duality defect of 2111.01141 is a twisted theta defect, but I was under the impression that this could also be obtained by composing interfaces (e.g. for $\mathcal{N}=4$ SYM at $\tau = i$, you compose the half-space gauging with a modular $\mathsf{S}$ transformation). Is this not what you had in mind?

---

## Round 2 · Author Response

We thank the referees for their careful reading of our paper and the comments and suggestions. Below is a response to the referees comments and a list of corresponding changes to the paper:
Report 1:
- The referee has queried whether the term "theta defects" is necessary or is simply a different terminology for condensation defects. Theta defects are {\it not} the same as condensation. E.g. for 2-group symmetries theta defects are shown to be distinct to condensation [ref 7 in the current version]. Other examples are twisted thetas, which are not condensation defects either. So introducing this terminology is crucial, and it unifies the perspective on non-invertible symmetries substantially: it incorporates condensation, but is much more than these. To stress this point, which is indeed important, we have added a box at the end of section 2 to emphasis this distinction. We hope this clarifies this point.
- We illustrated in detail in our companion paper https://arxiv.org/pdf/2212.06842.pdf how the mathematical formalism of this paper can be used to perform concrete computations.
- We thank the referee for pointing this out. This has now been referred to on top of page 19.
- As mentioned in the introduction, it is indeed left to future work, but this does not imply that the program discussed in section 4 is only for non-intrinsic symmetries. The particular aspect of the program that might connect to intrinsic symmetries is the one related to interfaces discussed in section 2.4.
Report 2:
We have corrected the typos pointed out by the referee and provided additional information according to referee's comments. Regarding point 4, what referee points out in the N=4 example is indeed correct, but it is not clear to us that the two types of defects are the same in general, and it is for this reason that we treat the two types separately.

You are currently on this page

---

## Editorial Decision

published